# Rate-Optimal Subspace Estimation on Random Graphs

**Zhixin Zhou[1], Fan Zhou[2], Ping Li[2], and Cun-Hui Zhang[3]**

[1]Department of Management Sciences, City University of Hong Kong
[2]Cognitive Computing Lab, Baidu Research
[3]Department of Statistics, Rutgers University
[1]zhixzhou@cityu.edu.hk, [2]{zfyyde001, pingli98}@gmail.com, [3]cunhui@stat.rutgers.edu

## Abstract

We study the theory of random bipartite graph whose adjacency matrix is generated according to a connectivity matrix $\mathbf{M}$. We consider the bipartite graph to be sparse, i.e., the entries of $\mathbf{M}$ are upper bounded by certain sparsity parameter. We show that the performance of estimating the connectivity matrix $\mathbf{M}$ depends on the sparsity of the graph. We focus on two measurement of performance of estimation: the error of estimating $\mathbf{M}$ and the error of estimating the column space of $\mathbf{M}$. In the first case, we consider the operator norm and Frobenius norm of the difference between the estimation and the true connectivity matrix. In the second case, the performance will be measured by the difference between the estimated projection matrix and the true projection matrix in operator norm and Frobenius norm. We will show that the estimators we propose achieve the minimax optimal rate.

## 1 Introduction

There have been many fruitful results on subspace estimation principal component analysis. Researchers have been considering random matrix with Gaussian and sub-Gaussian noise. In the present work, we focus on random *bipartite graph*, which have been a popular model in machine learning and data mining [26, 11, 19]. A bipartite graph is a graph whose vertices can be divided into two disjoint sets, $\mathcal{U}$ and $\mathcal{V}$, such that every edge connects a vertex in $\mathcal{U}$ to one in $\mathcal{V}$. In a random bipartite graph, an edge connecting node $i \in \mathcal{U}$ to node $j \in \mathcal{V}$ exists with probability $\mathbf{M}_{ij}$ independently. Suppose $|\mathcal{U}| = n_1$ and $|\mathcal{V}| = n_2$, a bipartite graph can be represented by an *adjacency matrix* $\mathbf{A}$, whose elements $\mathbf{A}_{ij}$, $(i, j) \in [n_1] \times [n_2]$ are independent Bernoulli random variables with latent parameters $\mathbf{M}_{ij}$. In other words, we can focus on the random adjacency matrix $\mathbf{A}$ with entries

$$\mathbf{A}_{ij} \sim \text{Bernoulli}(\mathbf{M}_{ij}) \quad \text{for } (i, j) \in [n_1] \times [n_2] \text{ independently.} \tag{1}$$

$\mathbf{M}$ with elements $\mathbf{M}_{ij}$ is called the *connectivity matrix*. The goal of this paper is to find the rate optimal estimation of the latent connectivity matrix $\mathbf{M}$ and its singular space, under the assumption that $\mathbf{M}$ follows certain low-rank structure.

Formally speaking, for the task of estimating matrix $\mathbf{M}$, we consider the following parameter space:

$$\boldsymbol{\Theta}_1(n_1, n_2, p, r) = \{\mathbf{M} \in [0, p]^{n_1 \times n_2} : \text{rank}(\mathbf{M}) \leq r\} \quad \text{for positive integers } n_1, n_2, r. \tag{2}$$

In other words, $\boldsymbol{\Theta}_1(n_1, n_2, p, r)$ consists of all $n_1 \times n_2$ matrices with entries between 0 and $p$ and rank at most $r$. To avoid empty parameter space, we always assume $r \leq \min\{n_1, n_2\}$. The parameter $p$ controls the sparsity of the bipartite graph.

The work of Zhixin Zhou was conducted as a postdoctoral researcher at Baidu Research - Bellevue WA.
The work of Cun-Hui Zhang was conducted as a consulting researcher at Baidu Research - Bellevue WA.

35th Conference on Neural Information Processing Systems (NeurIPS 2021).

Matrix estimation has been studied under different assumptions. In [7], the author assumes the random matrix has bounded entries. The universal singular value thresholding (USVT) algorithm is proved to be optimal in the parameter space with bounded nuclear norm. In [15], the authors consider a more general parameter space with subexponential random matrices with bounded means. However, their approach is not optimal under our random graph assumption. One-bit matrix completion is considered in [5, 9], but parameters therein are not necessarily of low rank matrices, so the minimax error is not comparable with that of the family (2) in the present paper. In network analysis problems, the density of network is expected to be sparse. The upper bound of connection probability $p$ in our model (2) plays a key role in error rates of the estimation.

When we consider the singular space estimation problem, an extra assumption about the smallest non-zero singular value is required. We define the corresponding parameter space as follows:

$$\boldsymbol{\Theta}_2(n_1, n_2, p, r, \sigma_*) = \{\mathbf{M} \in [0, p]^{n_1 \times n_2} : \text{rank}(\mathbf{M}) = r, \sigma_r(\mathbf{M}) \geq \sigma_*\}, \tag{3}$$

where $\sigma_r(\mathbf{M})$ is the $r$th largest singular value of $\mathbf{M}$. It is clear that $\boldsymbol{\Theta}_2(n_1, n_2, p, r, \sigma_*) \subset \boldsymbol{\Theta}_1(n_1, n_2, p, r)$. In other words, in the singular space estimation problem, we consider parameters in $\boldsymbol{\Theta}_1(n_1, n_2, p, r)$ whose smallest nonzero singular value is bounded below by $\sigma_*$.

Singular space estimation in the Gaussian case has been considered in [4]. To show the optimality of their estimation, they use packing number of Grassmanian manifold provided in [21]. In the random bipartite graph model that we consider in this paper, the connectivity matrix $\mathbf{M}$ in (3) is a merely proper subset of the Grassmanian manifold because the entries of $\mathbf{M}$ are contained in the interval $[0, p]$. As a consequence, the previous approach fails to calculate the lower bound of the packing number of the parameter space $\boldsymbol{\Theta}_2(n_1, n_2, p, r, \sigma_*)$. In the literature of community detection, spectral methods have been applied in clustering tasks on random graphs [17, 8, 13, 28, 29, 30]. The authors in the paper [27] consider this problem under information diffusion. However, these works have not focused on the optimality of singular space estimation. With the help of recent development in non-asymptotic analysis of random matrices, we propose a stochastic approach to compute the minimax lower bound of subspace estimation over the parameter space (3).

Here are some notations we will use in this paper. For real numbers $a$ and $b$, $a \vee b = \max\{a, b\}$ and $a \wedge b = \min\{a, b\}$. $\sigma_r(\mathbf{M})$ is the $r$th largest singular value of matrix $\mathbf{M}$. $\|\mathbf{M}\|_{\text{op}} = \sigma_1(\mathbf{M})$, i.e., the largest singular value, is the spectral norm of $\mathbf{M}$. $\|\mathbf{M}\|_{\text{F}} = \left(\sum_{i=1}^{n_1} \sum_{j=1}^{n_2} \mathbf{M}_{ij}^2\right)^{1/2}$ is the Frobenius norm of $\mathbf{M}$. We note that $\|\mathbf{M}\|_{\text{F}}^2 = \sum_{i=1}^{n_1 \wedge n_2} \sigma_i(\mathbf{M})^2$. $\text{Col}(\mathbf{M})$ is the column space of $\mathbf{M}$. Suppose $\mathbf{a} = (a_1, a_2, \ldots, a_n) \in \mathbb{R}^n$, then $\text{diag}(\mathbf{a})$ is an $n \times n$ diagonal matrix with entries $a_1, a_2, \ldots, a_n$ on the diagonal. For positive integer $n$, $[n] := \{1, 2, \ldots, n\}$. For positive sequence $a_n$ and $b_n$, if there exists a constant $C$ such that $a_n \leq Cb_n$ for all $n$, then we write $a_n = O(b_n)$ or $a_n \lesssim b_n$. If $a_n = O(b_n)$ and $b_n = O(a_n)$, then we write $a_n = \Theta(b_n)$. If $a_n/b_n \to 0$ as $n \to \infty$, then we write $a_n = o(b_n)$ or $a_n \ll b_n$. When we discuss asymptotic properties, we consider $\min\{n_1, n_2\} \to \infty$, i.e., the size the bipartite graph tends to infinity.

The rest of this paper is organized as follows. In Section 2, we introduce the algorithm for matrix estimation problem and also provide theoretical analysis for the error rate and minimax lower bound of this problem. Applying the result of matrix estimation, we propose the algorithm for column and row space estimation in Section 3. Then we show that the algorithm outputs estimator with minimax optimal error rate. Numerical experiments on the algorithms appear in Section 4. We discuss some possible future works in Section 5. The proofs of the theorems can be found in the supplementary material.

## 2 Low Rank Matrix Estimation of Random Graph

Given a random bipartite graph, we aim to estimate true connecting probability of the adjacency matrix assuming that the connectivity matrix has low rank. To be more specific, in this section, we consider the task of low rank matrix estimation parameters belong to the space

$$\boldsymbol{\Theta}_1(n_1, n_2, p, r) = \{\mathbf{M} \in [0, p]^{n_1 \times n_2} : \text{rank}(\mathbf{M}) \leq r\}$$

from (2). Without loss of generality, we assume $n_1 \geq n_2$ in this section. We recall that a random bipartite graph with adjacency matrix $\mathbf{A}$ is observed, and $\mathbf{A}_{ij} \sim \text{Bernoulli}(\mathbf{M}_{ij})$ independently for $(i, j) \in [n_1] \times [n_2]$. We will first proposed the algorithm to estimate $\mathbf{M}$, then we will show the optimality of this estimator in the minimax sense.

## 2.1 Hard Singular Value Thresholding

We first introduce the algorithm to compute the estimator $\hat{\mathbf{M}}$. The hard thresholding procedure is similar as universal singular value thresholding proposed in [7]. However, our algorithm is specifically designed for random graph, which leads to tighter error rate. The algorithm first applies degree truncation on bipartite adjacency matrix $\mathbf{A}$, then compute the best low rank approximation on the resulting matrix by singular value decomposition (SVD). For $n_1 \times n_2$ matrix $\mathbf{X}$, and recall that we assume $n_1 \geq n_2$, the SVD of $\mathbf{X}$ is defined by

$$\mathbf{X} = \sum_{i=1}^{n_2} \sigma_i \mathbf{U}_i \mathbf{V}_i^\top = \mathbf{U}\boldsymbol{\Lambda}\mathbf{V}^\top$$

where

$$\mathbf{U} = (\mathbf{U}_1, \mathbf{U}_2, \ldots, \mathbf{U}_{n_2}), \quad \boldsymbol{\Lambda} = \mathrm{diag}(\sigma_1, \sigma_2, \ldots, \sigma_{n_2}), \quad \text{and} \quad \mathbf{V} = (\mathbf{V}_1, \mathbf{V}_2, \ldots, \mathbf{V}_{n_2})$$

with singular values $\sigma_1 \geq \sigma_2 \geq \cdots \geq \sigma_{n_2}$. Now we are ready to present Algorithm 1 for estimating $\mathbf{M}$ by hard singular value thresholding.

---

**Algorithm 1** Hard Singular Value Thresholding

---

1: **Input:** adjacency matrix $\mathbf{A}$, sparsity parameter $p$, rank $r$.
2: **Output:** the estimate of $\mathbf{M}$, $\hat{\mathbf{M}}$.
3: Set $r' = r \wedge \lfloor n_2 p \rfloor$.
4: If the degree of any row or column of $\mathbf{A}$ is greater than $2n_1p$, replace a subset of 1's in such rows and columns by 0 so that the column-degrees and the row degrees of the resulting matrix are bounded by $2n_1p$. We call this regularized matrix $\mathbf{A}_{\mathrm{re}}$.
5: Let $\hat{\mathbf{M}}$ be the best rank-$r'$ approximation of $\mathbf{A}_{\mathrm{re}}$ composed of the first $r'$ SVD components with the largest singular values. (More details of this step will be discussed later in this section. )

---

**Remark 1.** *Suppose the rank of the connectivity matrix $\mathbf{M}$ is as large as the maximal expected row degree $n_2p$, then it is not sufficient to propose a low rank approximation. In this case, the rate of the estimation error will be the same as a trivial estimation with $\hat{\mathbf{M}} = \mathbf{A}$. Step 3 in this algorithm plays role as checking if $r$ is sufficiently small.*

**Remark 2.** *We note that the regularization in step 4 is necessary to achieve the optimal error rate. See experimental and theoretical result in [16]. It is worth noting that the regularization method presented in the algorithm is not the unique one. Besides replacing the entries by zeroes, we can multiply the columns and rows by a small constant so that the degree is bounded by $2n_1p$. Applying this method, the regularized adjacency matrix loses less information, so the estimation is more accurate. The threshold of regularization can be replaced by $cn_1p$ rather than $2n_1p$. In most cases, the best $c$ can be smaller than 2, though the theoretical estimation error rate cannot be improved by choosing the optimal $c$. We will discuss the choice of $c$ by simulation in Section 4.*

We will describe more details in finding best rank-$r'$ approximation in step 5. Suppose the regularized adjacency matrix $\mathbf{A}_{\mathrm{re}}$ from step 4 has SVD

$$\mathbf{A}_{\mathrm{re}} = \sum_{i=1}^{n_2} \sigma_i \mathbf{U}_i \mathbf{V}_i^\top,$$

with singular values $\sigma_1 \geq \sigma_2 \geq \cdots \geq \sigma_{n_2}$, then we define the estimator $\hat{\mathbf{M}}$ to be the best rank-$r'$ approximation $\mathbf{A}_{\mathrm{re}}$:

$$\hat{\mathbf{M}} = \sum_{i=1}^{r'} \sigma_i \mathbf{U}_i \mathbf{V}_i^\top.$$

We note that $\hat{\mathbf{M}}$ does not necessarily belong to the parameter space $\boldsymbol{\Theta}_1(n_1, n_2, p, r)$, but there is no further operation that can improve the error rate in the next theorem.

Now we are ready to present the high probability bounds for estimation of $\mathbf{M}$. The error will be measured by both operator norm $\|\hat{\mathbf{M}} - \mathbf{M}\|_{\mathrm{op}}$ and Frobenius norm $\|\hat{\mathbf{M}} - \mathbf{M}\|_{\mathrm{F}}$. The upper bounds of the error rate are shown in the following theorem.

**Theorem 1.** *Given the adjacency matrix $\mathbf{A}$ generated by parameter $\mathbf{M}$ from space $\boldsymbol{\Theta}_1(n_1, n_2, p, r)$ and assuming $n_1 \geq n_2$, the estimator $\hat{\mathbf{M}}$ obtained from Algorithm 1 satisfies*

$$\sup_{\mathbf{M}\in\boldsymbol{\Theta}_1(n_1,n_2,p,r)} \mathbb{P}\big(\|\hat{\mathbf{M}} - \mathbf{M}\|_{op}^2 \gtrsim (n_1 p) \wedge (n_1 n_2 p^2)\big) \leq n_1^{-1}; \tag{4}$$

$$\sup_{\mathbf{M}\in\boldsymbol{\Theta}_1(n_1,n_2,p,r)} \mathbb{P}\big(\|\hat{\mathbf{M}} - \mathbf{M}\|_F^2 \gtrsim (n_1 pr) \wedge (n_1 n_2 p^2)\big) \leq n_1^{-1}. \tag{5}$$

This theorem shows that if we consider the average loss of estimating entries of $\mathbf{M}$,

$$\frac{1}{n_1 n_2}\|\hat{\mathbf{M}} - \mathbf{M}\|_F^2 \lesssim \frac{pr}{n_2} \wedge p^2$$

holds with high probability as long as the dimension of the matrix tends to infinity. $p^2$ is a trivial upper bound since this can be obtained by letting $\hat{\mathbf{M}} = 0$. As long as $r = o(n_2 p)$, the low rank structure of $\mathbf{M}$ leads to a better bound in entrywise loss. This coincides with the discussion in Remark 1.

This upper bound has also been studied in [13, 28]. The authors applied their result for community detection when the number of communities are sufficiently smaller than the average degree of the graph. However, the term $n_1 n_2 p^2$ is missing in previous works.

**Remark 3.** *Without the Step 4 in Algorithm 1, (4) can still achieve the error rate $[(n_1 p) \vee (\log n_1)] \wedge (n_1 n_2 p^2)$ with high probability. One can check the technical reason from the concentration of random graph without regularization in [10, 17, 16]. However, this error rate cannot achieve the minimax lower bound in Theorem 3.*

## 2.2 Soft Singular Value Thresholding

Soft singular value thresholding [3, 15, 6] can also be applied to estimate the connectivity matrix $\mathbf{M}$. Hard thresholding truncates the smallest $n - r'$ singular value to 0, where $r'$ is defined in step 3 of Algorithm 1. In the soft thresholding approach, Algorithm 2 shrinks all singular values towards zero.

---

**Algorithm 2** Soft Singular Value Thresholding

1: **Input:** adjacency matrix $\mathbf{A}$, sparsity parameter $p$, rank $r$.
2: **Output:** the estimate of $\mathbf{M}$, $\hat{\mathbf{M}}$.
3: Set $r' = r \wedge \lfloor n_2 p \rfloor$.
4: Apply regularization (step 4 of Algorithm 1) on $\mathbf{A}$ to obtain $\mathbf{A}_{\text{re}}$.
5: Let $\mathbf{A}_{\text{re}} = \mathbf{U}\boldsymbol{\Lambda}\mathbf{V}^\top$ be the SVD of $\mathbf{A}_{\text{re}}$ where $\boldsymbol{\Lambda} = \text{diag}(\sigma_1, \sigma_2, \ldots, \sigma_{n_2})$.
6: Let $\hat{\boldsymbol{\Lambda}} = \text{diag}((\sigma_1 - \sigma_{r'+1}) \vee 0, (\sigma_2 - \sigma_{r'+1}) \vee 0, \ldots, (\sigma_{n_1 \wedge n_2} - \sigma_{r'+1}) \vee 0)$.
7: Let $\hat{\mathbf{M}} = \mathbf{U}\hat{\boldsymbol{\Lambda}}\mathbf{V}^\top$.

---

**Theorem 2.** *Under the same assumptions in Theorem 1, the estimator $\hat{\mathbf{M}}$ obtained from Algorithm 2 satisfies (4) and (5).*

Theorem 1 and Theorem 2 show that both hard and soft singular value thresholding methods provide consistent estimation of $\mathbf{M}$ up to a constant factor. Essentially, the estimator $\hat{\mathbf{M}}$ in each algorithm is a low rank matrix satisfying $\|\mathbf{M} - \hat{\mathbf{M}}\|_{\text{op}}^2 \lesssim (n_1 p) \wedge (n_1 n_2 p^2)$ with high probability.

## 2.3 Minimax Lower Bound

Theorem 1 and Theorem 2 provide a high-probability upper bound for the estimation in $\mathbf{M}$. We now establish lower bounds for the estimation error measured by operator norm and Frobenius distance. We will show that the high probability bounds in (4) and (5) are minimax optimal in the next theorem.

**Theorem 3.** *Suppose $n_1 \geq n_2$ and $n_1$ is sufficiently large, then*

$$\inf_{\hat{\mathbf{M}}} \sup_{\mathbf{M}\in\boldsymbol{\Theta}_1(n_1,n_2,p,r)} \mathbb{P}\Big(\|\hat{\mathbf{M}} - \mathbf{M}\|_{op}^2 \gtrsim (n_1 p) \wedge (n_1 n_2 p^2)\Big) \geq \frac{1}{2}; \tag{6}$$

$$\inf_{\hat{\mathbf{M}}} \sup_{\mathbf{M}\in\boldsymbol{\Theta}_1(n_1,n_2,p,r)} \mathbb{P}\Big(\|\hat{\mathbf{M}} - \mathbf{M}\|_F^2 \gtrsim (n_1 pr) \wedge (n_1 n_2 p^2)\Big) \geq \frac{1}{2}. \tag{7}$$

Here, the infimum is over all estimators depending on $\mathbf{A}, p$ and $r$. (6) shows that there is no estimator can achieve a high probability bound better than $\Theta((n_1 p) \wedge (n_1 n_2 p^2))$ when we measure the estimation error by operator norm. This lower bound can be extended to random tensor [31]. Similarly, (7) shows the minimax optimality in Frobenius norm. We note that the numerical constants hidden in the notation $\gtrsim$ is smaller the ones hidden in the notation $\lesssim$ in (4) and (5).

These lower bounds are comparable with [15, Theorem 5]. Although in their paper, the authors consider Gaussian noise rather than Bernoulli random variables, the construction of packing number in the proofs are similar. Nevertheless, the upper bound and lower bound in their paper does not match if the entrywise standard deviation and $\max_{i,j} |M_{ij}|$ are not asymptotically equivalent. In the situation of random graph, we assume that $\max_{i,j} |M_{ij}| = p$, then $p \ll \sqrt{p(1-p)}$ if the random graph is sparse, i.e., $p \ll 1$. In this case, the minimax lower bound in [15] is suboptimal. Theorem 3 resolves this issue and show that Algorithm 1 and Algorithm 2 return optimal estimator in the random bipartite graph problem.

## 2.4 Comparison with Stochastic Block Models and Graphon Models

Popular models on random graphs including stochastic block models (SBM) and graphon models have been investigated in previous work. There have been wonderful results on these models in literature. We will discuss the relationship of the previous results with our current work.

In SBM, the probability of the existence of an edge only depends on the membership of its incident nodes. We consider the SBM on bipartite graph in [29]. Suppose there are $K$ communities in $\mathcal{U}$ and $L$ communities in $\mathcal{V}$, then the connectivity matrix $\mathbf{M}$ consists of at most $KL$ many distinct entries. The rank of $\mathbf{M}$ is at most $K \wedge L$. To simplify the discussion, we assume $r = K = L$. SBM can be considered as a low-rank model, but the parameter space of SBM is much smaller than $\boldsymbol{\Theta}_1(n_1, n_2, p, r)$. There are about $(n_1 + n_2)r$ parameters in the low-rank model, while there are only $r^2$ parameters in SBM, i.e., one parameter for each block. Given the size of the random bipartite graph is the same, on average, there are less random Bernoulli observations for estimating each parameter in the low-rank model. Therefore, the error rate in matrix estimation on SBM [12, 14] is tighter than low-rank model. Since SBM is a low-rank model, spectral methods have been applied to SBM. The spectral clustering on SBM typically applies $K$-means type algorithms on $\hat{\mathbf{M}}$ to obtain consistent community labels [20, 8, 25, 17, 13, 28].

The parameter space of graphon models is characterized by smoothness conditions on a continuous function. The details can be found in [12, 14]. In these papers, the authors use the connectivity matrix in SBM to estimate the function in the graphon model. The number of the blocks in SBM depends on the smoothness assumption of the graphon. It is worth noting that estimating graphon by a connectivity matrix in SBM is not computationally feasible. In another paper [23], spectral methods are applied to graphon estimation, but they fail to achieve the optimal rate.

# 3 Singular Space Estimation

In this section, we will study the estimation of column and row spaces of parameter $\mathbf{M}$. This task is closely related to the estimation of $\mathbf{M}$ which has been studied in the previous section. Compared with the parameter space $\boldsymbol{\Theta}_1(n_1, n_2, p, r)$, it additionally requires $\sigma_r(\mathbf{M})$ to be bounded away from zero, where $\sigma_r(\mathbf{M})$ is the $r$th largest singular value of $\mathbf{M}$. This is a necessary condition to distinguish the subspace associated with top singular values from its normal space. We recall the parameter space from (3):

$$\boldsymbol{\Theta}_2(n_1, n_2, p, r, \sigma_*) = \{\mathbf{M} \in [0, p]^{n_1 \times n_2} : \text{rank}(\mathbf{M}) = r, \sigma_r(\mathbf{M}) \geq \sigma_*\}.$$

For $\mathbf{M} \in \boldsymbol{\Theta}_2(n_1, n_2, p, r, \sigma_*)$, $\mathbf{M}^\top \mathbf{M} \in [0, n_2 p^2]^{n_1 \times n_1}$. Therefore,

$$n_1 n_2 p^2 \geq \text{tr}(\mathbf{M}^\top \mathbf{M}) = \sum_{i=1}^{n_1 \wedge n_2} \sigma_i^2(\mathbf{M}) \geq \sigma_*^2 r. \tag{8}$$

Hence, $\sigma_* \leq \sqrt{\frac{n_1 n_2}{r}} p$ is a necessary condition for the parameter space $\boldsymbol{\Theta}_2(n_1, n_2, p, r, \sigma_*)$ being nonempty. Recall that we assume the observed bipartite adjacency matrix follows the distribution

$$\mathbf{A}_{ij} \sim \text{Bernoulli}(\mathbf{M}_{ij}) \quad \text{for } (i, j) \in [n_1] \times [n_2] \text{ independently.}$$

We aim to recover the top-$r$ left and right singular vectors. Due to the symmetry of the column and row space, we will focus on left singular vectors in the following discussion. Let

$$\mathbf{M} = \sum_{i=1}^{r} \sigma_i(\mathbf{M})\mathbf{U}_i\mathbf{V}_i^\top$$

be the singular value decomposition of $\mathbf{M}$. Our goal is to recover the subspace spanned by $\{\mathbf{U}_1, \ldots, \mathbf{U}_r\}$. Let $\mathbf{U} = (\mathbf{U}_1, \ldots, \mathbf{U}_r)$. Since the columns of $\mathbf{U}$ are singular vectors of $\mathbf{M}$, $\mathbf{U}$ has orthogonal columns. Suppose the estimator $\hat{\mathbf{U}}$ also has orthogonal columns, then the projection matrices onto columns spaces of $\mathbf{U}$ and $\hat{\mathbf{U}}$ are $\mathbf{U}\mathbf{U}^\top$ and $\hat{\mathbf{U}}\hat{\mathbf{U}}^\top$ respectively. The loss in operator norm and Frobenius norm are defined by

$$L_{\mathrm{op}}(\hat{\mathbf{U}}, \mathbf{U}) = \|\hat{\mathbf{U}}\hat{\mathbf{U}}^\top - \mathbf{U}\mathbf{U}^\top\|_{\mathrm{op}}^2 \quad \text{and} \quad L_{\mathrm{F}}(\hat{\mathbf{U}}, \mathbf{U}) = \|\hat{\mathbf{U}}\hat{\mathbf{U}}^\top - \mathbf{U}\mathbf{U}^\top\|_{\mathrm{F}}^2. \tag{9}$$

We can similarly define the loss of row space estimator by $L_{\mathrm{op}}(\hat{\mathbf{V}}, \mathbf{V})$ and $L_{\mathrm{F}}(\hat{\mathbf{V}}, \mathbf{V})$.

**Remark 4.** *Due to the identifiability issue, we cannot directly compare the true basis of the column space of $\mathbf{M}$ with the estimator $\hat{\mathbf{U}}$. In this paper, we use the difference between the projection matrices corresponding to different subspaces to measure the estimation error. There are other norms using in literature, for instance, the $\sin\Theta$ distance*

$$\|\sin\Theta(\hat{\mathbf{U}}, \mathbf{U})\| = \|\mathbf{U}^\top\hat{\mathbf{U}}_\perp\|$$

*where $\hat{\mathbf{U}}_\perp$ has orthonormal columns which spans the normal space of $\mathrm{Col}(\hat{\mathbf{U}})$. Another possible measurement of distance between $\mathbf{U}$ and $\hat{\mathbf{U}}$ is*

$$\inf_{\mathbf{O}\in\mathbb{O}_r} \|\mathbf{U} - \hat{\mathbf{U}}\mathbf{O}\|.$$

*Here, the norm $\|\cdot\|$ can be either the operator norm or the Frobenius norm, and $\mathbb{O}_r$ represents the set of $r \times r$ orthogonal matrix. It is shown in [4] that these two norms are equivalent to the metric on the difference between projection matrices defined in (9).*

### 3.1 Singular Subspace Estimation and Upper Bound

We consider the rank truncation algorithm. In Algorithm 3, we first obtain the estimator $\hat{\mathbf{M}}$ from Algorithm 1 or Algorithm 2, then we collect the top singular vectors of $\hat{\mathbf{M}}$. The subspace spanned by these singular vectors is the estimation of the column space of $\mathbf{M}$. Similarly ,the algorithm can output the row space of $\mathbf{M}$.

---

**Algorithm 3** Singular space estimation

---

1: **Input:** adjacency matrix $\mathbf{A}$, sparsity parameter $p$, rank $r$.
2: **Output:** the estimate of the column space and row space of $\mathbf{M}$.
3: Let $\hat{\mathbf{M}}$ be the output of Algorithm 1 or Algorithm 2.
4: Let $\hat{\mathbf{U}} = (\mathbf{u}_1, \ldots, \mathbf{u}_r)$ where $\mathbf{u}_i$'s are the left singular vectors of $\hat{\mathbf{M}}$ corresponding to top-$r$ singular values.
5: Let $\hat{\mathbf{V}} = (\mathbf{v}_1, \ldots, \mathbf{v}_r)$ where $\mathbf{v}_i$'s are the right singular vectors of $\hat{\mathbf{M}}$ corresponding to top-$r$ singular values.
6: Output $\hat{\mathbf{U}}$ and $\hat{\mathbf{V}}$.

---

We combine the upper bound in Theorem 1 with Davis-Kahan theorem, then obtain the upper bound of loss in subspace estimation given by Algorithm 3.

**Theorem 4.** *Given the adjacency matrix $\mathbf{A}$ generated by connectivity matrix $\mathbf{M}$ from parameter space $\mathbf{\Theta}_2(n_1, n_2, p, r, \sigma_*)$, the outputs of $\hat{\mathbf{U}}$ and $\hat{\mathbf{V}}$ from Algorithm 3 satisfy*

$$\sup_{\mathbf{M}\in\mathbf{\Theta}_2(n_1,n_2,p,r,\sigma_*)} \mathbb{P}\left( \max(L_{op}(\hat{\mathbf{U}}, \mathbf{U}), L_{op}(\hat{\mathbf{V}}, \mathbf{V})) \gtrsim \frac{(n_1 + n_2)p}{\sigma_*^2} \wedge 1 \right) \leq (n_1 \vee n_2)^{-1}; \tag{10}$$

$$\sup_{\mathbf{M}\in\mathbf{\Theta}_2(n_1,n_2,p,r,\sigma_*)} \mathbb{P}\left( \max(L_F(\hat{\mathbf{U}}, \mathbf{U}), L_F(\hat{\mathbf{V}}, \mathbf{V})) \gtrsim \frac{(n_1 + n_2)pr}{\sigma_*^2} \wedge r \right) \leq (n_1 \vee n_2)^{-1}. \tag{11}$$

The upper bound shows that if we obtain better a estimation $\hat{\mathbf{M}}$ or if the spectral gap $\sigma^*$ is larger, then we can get more accurate estimation in singular space of $\mathbf{M}$. Since the operator norm of a projection matrix is 1, the estimation error in operator norm cannot exceed 2. Therefore, 2 is a trivial upper bound (10). Similarly, $2r$ is a trivial upper bound in (11).

## 3.2 Minimax Lower Bound

In this section, we introduce the lower bound for the estimation in Theorem 4. Let us denote the projection matrix of the left singular space of $\mathbf{M}$ by

$$\mathbf{P}_{\mathbf{M}} = \mathbf{U}\mathbf{U}^\top.$$

where the columns of $\mathbf{U}$ are top-$r$ singular vector of $\mathbf{M}$. In the next theorem, we will show that under regularity conditions on the parameter space, the estimation error rate of $\mathbf{P}_{\mathbf{M}}$ in the previous theorem is minimax optimal.

**Theorem 5.** *Let $1 \leq r < n_2 \leq n_1$. Suppose that for some positive constant $C_0$, and assume*

*(a) $n_1 n_2 (4p/10)^2 \leq r\sigma_*^2 \leq n_1 n_2 (6p/10)^2$ when $r = 1$;*

*(b) $r \leq n_1/C_0$, $r\sigma_*^2 \leq n_1 n_2 p^2/C_0$ when $r > 1$.*

*Then, for sufficiently large $C_0$ there exist a prior $\pi^*$ such that*

$$\mathbb{P}_{\pi^*}\left\{\mathbf{M} \in [0.3p, 0.7p]^{n_1 \times n_2}, \sigma_r(\mathbf{M}) \geq \sigma_*, rank(\mathbf{M}) = r\right\} = 1 \tag{12}$$

*and the risk of the Bayes estimator $\hat{\mathbf{P}}^* = \mathbb{E}_{\pi^*}\left[\mathbf{P}_{\mathbf{M}}\big|\mathbf{A}\right]$ under $\mathbb{P}_{\pi^*}$ is bounded from below by*

$$\mathbb{E}_{\pi^*}\left[\|\hat{\mathbf{P}}^* - \mathbf{P}_{\mathbf{M}}\|_{op}^2\right] \geq \frac{1}{50}\left(\frac{n_1 p}{\sigma_*^2} \wedge 1\right) \tag{13}$$

*and*

$$\mathbb{E}_{\pi^*}\left[\|\hat{\mathbf{P}}^* - \mathbf{P}_{\mathbf{M}}\|_F^2\right] \geq \frac{r}{50}\left(\frac{n_1 p}{\sigma_*^2} \wedge 1\right). \tag{14}$$

**Remark 5.** *The assumptions on the parameters are mild and necessary (up to the choice of constants). $r/n_1$ has to be sufficiently small so that the parameter $\mathbf{M}$ is low-rank. In (8), we have shown that $n_1 n_2 p^2/(r\sigma_*^2)$ is sufficiently large so that the parameter space is nonempty. It requires the parameter space to be large enough to construct the prior $\pi_*$. We need slightly different regularity conditions for $r = 1$ and $r > 1$ because of the different construction of $\pi_*$ in each case.*

Theorem 5 presents the lower bound of the Bayes error rate. By the fact that a minimax estimator should be a Bayes estimator with respect to the least favorable prior of the parameter space, it is straightforward to obtain

$$\inf_{\hat{\mathbf{P}}} \sup_{\mathbf{M} \in \Theta_2(n_1, n_2, p, r, \sigma_*)} \mathbb{E}\left[\|\hat{\mathbf{P}} - \mathbf{P}_{\mathbf{M}}\|_{op}^2\right] \geq \frac{1}{50}\left(\frac{n_1 p}{\sigma_*^2} \wedge 1\right)$$

and

$$\inf_{\hat{\mathbf{P}}} \sup_{\mathbf{M} \in \Theta_2(n_1, n_2, p, r, \sigma_*)} \mathbb{E}\left[\|\hat{\mathbf{P}} - \mathbf{P}_{\mathbf{M}}\|_F^2\right] \geq \frac{r}{50}\left(\frac{n_1 p}{\sigma_*^2} \wedge 1\right).$$

from (13) and (14). We have been focus on the lower bound of the left subspace estimation in the discussion so far. The result of Theorem 5 can be extended to the right subspace estimation. We skip the presentation of this similar result.

**Proof Sketch of Theorem 5**   We only discuss the case $r > 1$ here. The case $r = 1$ has a different but simpler proof. As in the proofs of many other minimax lower bounds, we firstly construct a finite subset of the parameter space. Let $\mathbf{H} \in [-\sqrt{3}, \sqrt{3}]^{n_1 \times (r-1)}$ such that $(\mathbf{H}, \mathbf{1}_{n_1})^\top(\mathbf{H}, \mathbf{1}_{n_1})/n_1 = \mathbf{I}_r$. Let $\mathbf{X}_i, i = 1, \ldots, N$, be distinct matrices in $\{-1, 1\}^{n_1 \times (r-1)}$, with $N = 2^{n_1(r-1)}$, $\mathbf{W}_i = \sqrt{1 - \mu^2}\mathbf{H} + \mu\mathbf{X}_i$ with $0 < \mu < 1$, and

$$\mathbf{M}_i = \frac{p}{2}\mathbf{1}_{n_1 \times n_2} + \frac{p}{10}(\mathbf{W}_i, -\mathbf{W}_i, \ldots, \mathbf{W}_i, -\mathbf{W}_i, \mathbf{O}), \tag{15}$$

where $(\mathbf{W}_i, -\mathbf{W}_i)$ is repeated $k$ times. This alternating structure can help to provide sufficiently large spectral gap. The parameters $\mu$ and $k$ can control the distinguishability between distinct $\mathbf{M}_i$ and $\mathbf{M}_j$. We require $\mathbf{X}_i$

(a) has almost orthogonal columns;

(b) has columns almost orthogonal to the columns of $(\mathbf{H}, \mathbf{1}_{n_1})$,

(c) is sufficiently different from $\mathbf{X}_j$ for $j \neq i$.

There are sufficient amount of such $\mathbf{X}_i$ when $n_1$ is large. We assign a uniform prior on the remaining $\mathbf{M}_i$. The minimax error is bounded below by the Bayesian error. In addition, the lower bound of $\|\hat{\mathbf{P}}_{\mathbf{M}} - \mathbf{P}_{\mathbf{M}_i}\|_F$ can be controlled by $\|\hat{\mathbf{X}} - \mathbf{X}_i\|_F$. Therefore, the problem can be reduced to find the lower bound on Bayes error for estimating $\mathbf{X}_i$. The Bayes error is lower bounded by the Hellinger distance between the probability space with parameter $\mathbf{M}_i$ and $\mathbf{M}_j$. Measuring the Hellinger distance between Bernoulli random variables becomes a standard problem.

# 4  Numerical Experiments

In previous sections, we have shown that the algorithms returns rate optimal estimation in different tasks. The choice of hyperparameter does not affect the error rate in the theorems, but it can change the performance of the algorithms in practice. In this section, each experiment will repeat 100 times. In each iteration, the randomization procedure follows these steps:

1. Randomly generate matrices $\mathbf{M}_1 \in \mathbb{R}^{n_1 \times (r-1)}$ $\mathbf{M}_2 \in \mathbb{R}^{(r-1) \times n_2}$ so that each entries follows normal distribution independently.

2. $\mathbf{M}_3 = \mathbf{M}_1 \mathbf{M}_2$ is an $n_1 \times n_2$ matrix with rank $r - 1$.

3. Scale $\mathbf{M}_3$ so that all entries belongs to the interval $[0, p]$. We note that this step increases the rank of the matrix by 1. The resulting rank-$r$ matrix is the connectivity matrix $\mathbf{M}$ in the current iteration.

4. Generate the adjacency matrix of the random bipartite graph with connectivity matrix $\mathbf{M}$.

## 4.1  Simulation on Matrix Estimation

Algorithm 1 and Algorithm 2 have the same theoretical error rate. However, their performances in simulation are significant different. The experimental results of both algorithms will appear in this section for comparison. We consider the following parameters in $\mathbf{\Theta}_1(n_1, n_2, r, p)$. $n_1 = n_2 = 1000, r = 3, p = 0.01, 0.03, 0.05$. In the following experiments, we vary the regularization constant $c$ from 0.2 to 1, where the default constant equals to 2 in Algorithm 1. When we apply regularization on the adjacency matrix, we scale the columns and the rows so that the maximal degree is at most $cn_1p$, as described in Remark 2.

From the simulation results in Figure 1, we can conclude following observations:

1. Compared with soft singular value thresholding in Algorithm 2, the performance of hard singular value thresholding in Algorithm 1 more heavily relies on the choice of regularization parameter. If we are able to choose a good regularization parameter, Algorithm 1 outperforms Algorithm 2 when the error is measured by operator norm. On the other hand, if we consider the Frobenius loss, Algorithm 2 performs relatively better. Suppose we are unable to choose a good regularization constant, Algorithm 2 performs better in most cases.

2. Since the density parameter $p = 0.01, 0.03$ and $0.05$ on different columns from left to right, we compare the plots in the first row and observe that regularization has smaller effect as the density of the random bipartite graph increases. This phenomenon indeed coincides with the theoretical background described in Remark 3. Even if we do not apply regularization on the adjacency matrix, the loss rate can be bounded above by $(n_1 p) \vee (\log n_1)$. As $p$ increases, $\log n_1$ can take on the role of the upper bound.

3. Regularization has little affect on soft singular value thresholding (Algorithm 2). When $p = 0.01$, regularization constant $= 0.8$ works slightly better than 0.7 and 0.9. For $p = 0.03$ and $0.05$, the performances are almost identical when the regularization constant is greater than 0.6. The performance of Algorithm 2 in this experiment is slightly different from our expectation, but it does not violate any theorems in the previous sections. It could be an interesting future work to explain why regularization does not substantially improve the soft singular value thresholding method.

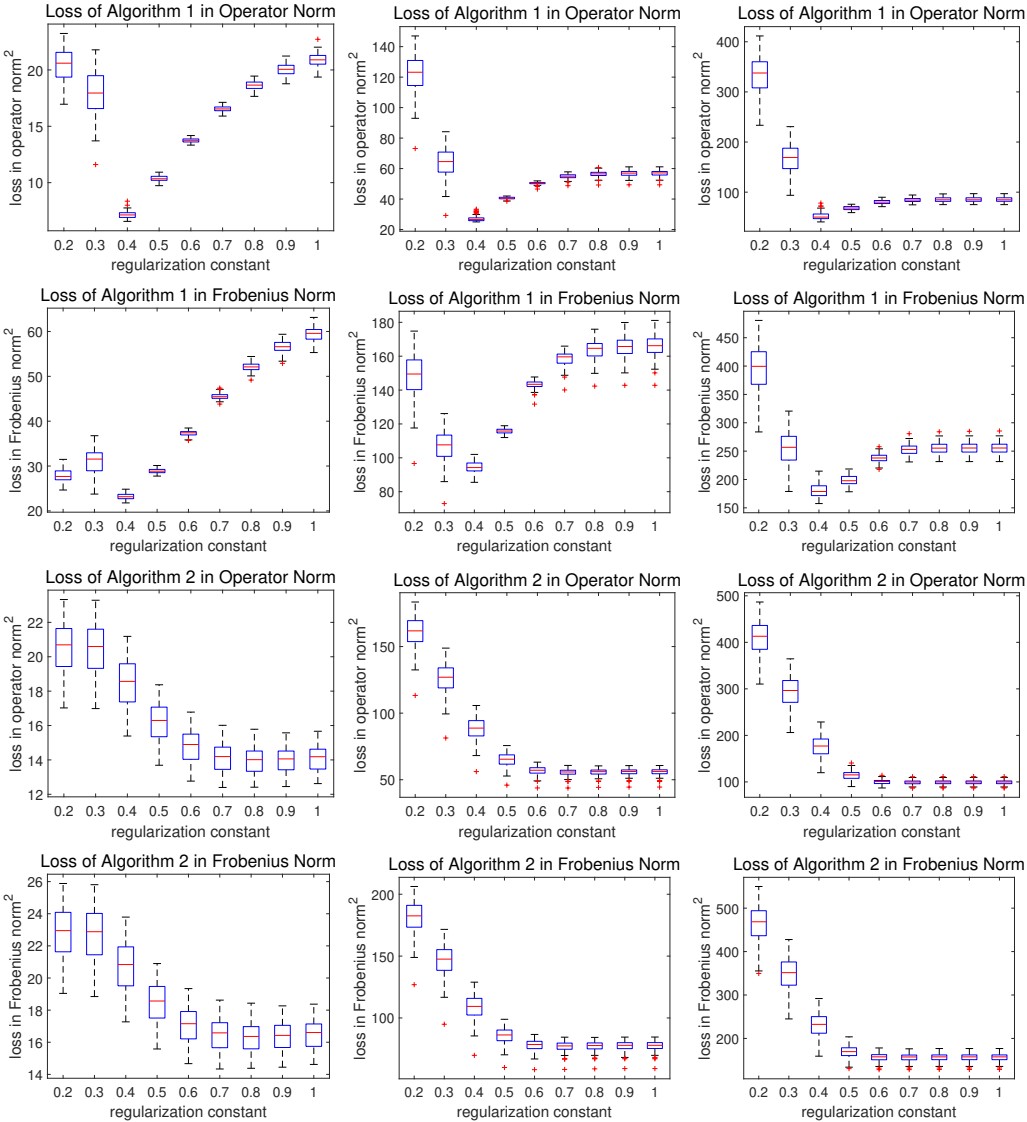

Figure 1: Numerical experiments of Algorithm 1 and Algorithm 2. The probability matrix $\mathbf{M}$'s are generated from the parameter space $\boldsymbol{\Theta}_1(n_1, n_2, r, p)$ with $n_1 = n_2 = 1000, r = 3$. $p = 0.01, 0.03, 0.05$ on different columns (from left to right). The horizontal axis represents the choice of regularization parameter.

## 4.2 Simulation on Subspace Estimation

The simulation results of Algorithm 3 are presented in Figure 2. The parameters in this experiment is the same as those used in Section 4.1. Due to the simulation procedure, it is difficult to control the smallest nonzero singular value of the connectivity matrix $\mathbf{M}$. Therefore, we plot the loss measured by $\sigma_*^2 L_{\mathrm{op}}(\hat{\mathbf{U}}, \mathbf{U})$ and $\sigma_*^2 L_{\mathrm{F}}(\hat{\mathbf{U}}, \mathbf{U})$. We observe that the regularization constant does not have a significant effect on the performance, but we can still conclude that the algorithm performs slightly better if we set the constant to be around 0.5 when we measure the loss in Frobenius norm.

## 5 Discussion

In this section, we will discuss some possible future works. We have proposed and analyzed the estimators given the parameters such as $p$ and $r$. In Section 3, we study the loss in estimating the column and row spaces simultaneously. We are wondering if we can extend these algorithms and error rates in more general settings.

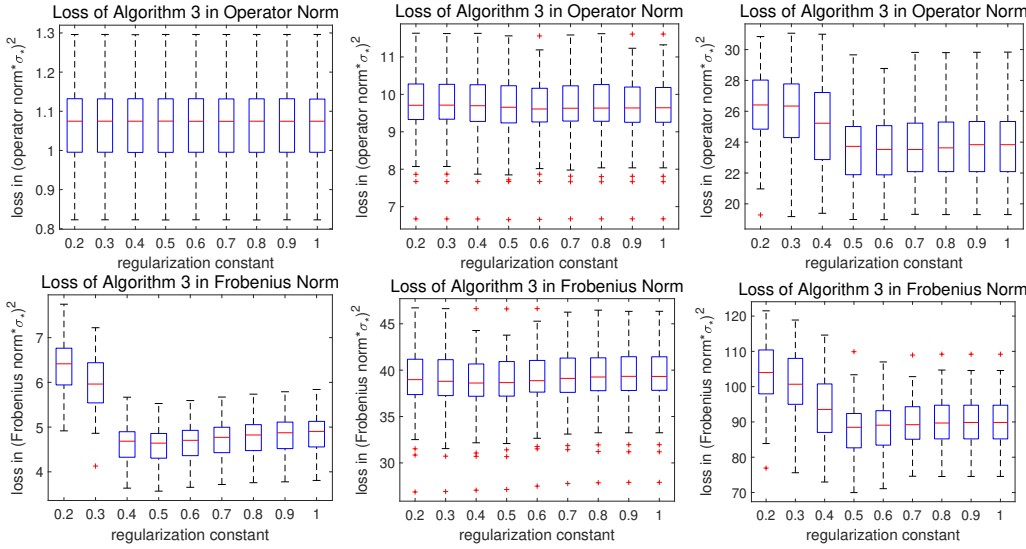

Figure 2: Numerical experiments of Algorithm 3. The probability matrix $\mathbf{M}$'s are generated from the parameter space $\mathbf{\Theta}_2(n_1, n_2, r, p, \sigma)$ with $n_1 = n_2 = 2000, r = 3$. $p = 0.01, 0.03, 0.05$ on different columns (from left to right). The horizontal axis represents the choice of regularization parameter.

## 5.1   Bipartite Graphs with Unbalanced Parts

Suppose the bipartite graphs has unbalanced parts, i.e., $n_1 \ll n_2$ for the dimension of the adjacency matrix, then the upper bound in Theorem 4 may not be optimal for $L_{\text{op}}(\mathbf{U}, \hat{\mathbf{U}})$. Intuitively, suppose there are more columns than rows in the adjacency matrix, then we are able to collect more information about the column space of the connectivity matrix $\mathbf{M}$. It is not clear how the assumption $n_1 \ll n_2$ affects the error rate of $L_{\text{op}}(\mathbf{U}, \hat{\mathbf{U}})$ in our theorem. On the other hand, increasing number of rows also changes the spectral gap $\sigma_*$. Some recent works [18, 2] have focused on unbalanced random matrix, but the authors have not shown their results are optimal in the random bipartite graph setting. The generalized theorem in [4] enable us to find a tight bound for random matrix with Gaussian entries, but the application on sparse random graphs is not straightforward. It is still an open problem to find the minimax error rate of $L_{\text{op}}(\mathbf{U}, \hat{\mathbf{U}})$ and $L_{\text{F}}(\mathbf{U}, \hat{\mathbf{U}})$.

## 5.2   Unspecified Parameters

To establish the minimax lower bound, we specify the sparsity parameter $p$ and the rank $r$. However, these parameters are unknown in general in practice. A possible approach is to tune these parameters by cross validation. We randomly partition the bipartite graph into a training subgraph and a validation subgraph, then we apply the algorithms in the present paper with various parameters and compute the likelihood on the validation subgraph. We can generalize this procedure to cross validation and find the parameters so that the average likelihood on validation subgraphs are maximized. The implementation of cross validation should be straightforward, while the risk of estimation without specified parameters can be an interesting future work.

# 6   Conclusion

In this paper, we study bipartite graphs with underlying low rank structures. We have introduced the algorithms for finding the connectivity matrix and its column and row space, using the techniques of regularization, hard and soft singular value thresholding. We provide theoretical analysis in Theorem 1, Theorem 2 and Theorem 4 for the algorithms we proposed. In addition, minimax lower bounds in Theorem 3 and Theorem 5 show that the algorithms return rate optimal estimators for corresponding tasks. In addition, simulation studies in Section 4 investigate the effect of regularization parameters in the algorithms. Finally, we discuss some possible future works in Section 5.

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
