# Rate-Optimal Subspace Estimation on Random Graphs

Zhixin Zhou[1], Fan Zhou[2], Ping Li[2], and Cun-Hui Zhang[3]

[1]Department of Management Sciences, City University of Hong Kong
[2]Cognitive Computing Lab, Baidu Research
[3]Department of Statistics, Rutgers University
[1]zhixzhou@cityu.edu.hk, [2]{zfyyde001, pingli98}@gmail.com, [3]cunhui@stat.rutgers.edu

## A    Proof of Theorem 1

**For operator norm.** Let $\hat{\mathbf{M}}$ be obtained from the last step of the algorithm, then by [16, Theorem 2.1], $\mathbf{A}_{\mathrm{re}}$ satisfies

$$\mathbb{P}(\|\mathbf{A}_{\mathrm{re}} - \mathbf{M}\|_{\mathrm{op}} \lesssim \sqrt{n_1 p}) \geq 1 - n_1^{-1}. \tag{16}$$

By triangle inequality, we have

$$\|\hat{\mathbf{M}} - \mathbf{M}\|_{\mathrm{op}} \leq \|\hat{\mathbf{M}} - \mathbf{A}_{\mathrm{re}}\|_{\mathrm{op}} + \|\mathbf{A}_{\mathrm{re}} - \mathbf{M}\|_{\mathrm{op}}. \tag{17}$$

Now it remains to find the upper bound for $\|\hat{\mathbf{M}} - \mathbf{A}_{\mathrm{re}}\|_{\mathrm{op}}$. We have

$$\mathbf{A}_{\mathrm{re}} - \hat{\mathbf{M}} = \sum_{i=1}^{n_2} \sigma_i(\mathbf{A}_{\mathrm{re}})\mathbf{U}_i\mathbf{V}_i^{\top} - \sum_{i=1}^{r'} \sigma_i(\mathbf{A}_{\mathrm{re}})\mathbf{U}_i\mathbf{V}_i^{\top} = \sum_{i=r'+1}^{n_2} \sigma_i(\mathbf{A}_{\mathrm{re}})\mathbf{U}\mathbf{V}^{\top}$$

Therefore, $\|\hat{\mathbf{M}} - \mathbf{A}_{\mathrm{re}}\|_{\mathrm{op}} = \sigma_{r'+1}(\mathbf{A}_{\mathrm{re}})$. Now it is sufficient to show that $\sigma_{r'+1}(\mathbf{A}_{\mathrm{re}}) \lesssim \sqrt{n_1 p}$ with high probability. Suppose $r' = r$, then $\sigma_{r'+1}(\mathbf{M}) = 0$. Suppose $r' = \lfloor n_2 p \rfloor$, then applying $\mathrm{tr}(\mathbf{M}^{\top}\mathbf{M}) \leq n_1 n_2 p^2$,

$$\sigma_{r'+1}(\mathbf{M}) \leq \sqrt{\frac{\mathrm{tr}(\mathbf{M}^{\top}\mathbf{M})}{r'+1}} \leq \sqrt{\frac{\mathrm{tr}(\mathbf{M}^{\top}\mathbf{M})}{n_2 p}} \leq \sqrt{n_1 p}. \tag{18}$$

By Weyl's inequality (Theorem 6), on the event $\|\mathbf{A}_{\mathrm{re}} - \mathbf{M}\|_{\mathrm{op}} \lesssim \sqrt{n_1 p}$,

$$\sigma_{r'+1}(\mathbf{A}_{\mathrm{re}}) \leq \sigma_{r'+1}(\mathbf{M}) + \|\mathbf{A}_{\mathrm{re}} - \mathbf{M}\|_{\mathrm{op}} \lesssim \sqrt{n_1 p}. \tag{19}$$

with probability at least $1 - n_1^{-1}$. This completes the proof for $\|\hat{\mathbf{M}} - \mathbf{M}\|_{\mathrm{op}} \lesssim \sqrt{n_1 p}$ with high probability. For $\|\hat{\mathbf{M}} - \mathbf{M}\|_{\mathrm{op}} \lesssim \sqrt{n_1 n_2 p^2}$, it is sufficient to show that $\|\hat{\mathbf{M}} - \mathbf{M}\|_{\mathrm{F}} \lesssim \sqrt{n_1 n_2 p^2}$. This will be proved as follows.

**For Frobenius norm.** Case 1: $r' = r$. Since $\hat{\mathbf{M}}$ and $\mathbf{M}$ has at most rank $r$, $\mathrm{rank}(\hat{\mathbf{M}} - \mathbf{M}) \leq 2r$. Thus,

$$\|\hat{\mathbf{M}} - \mathbf{M}\|_{\mathrm{F}} \leq \sqrt{2r}\|\hat{\mathbf{M}} - \mathbf{M}\|_{\mathrm{op}} \lesssim 2\sqrt{2n_1 p r},$$

which gives the desired result.
Case 2: $r' = \lfloor n_2 p \rfloor$. Let

$$\mathcal{T}_{r'}(\mathbf{M}) = \sum_{i=1}^{r'} \sigma_i(\mathbf{M})\mathbf{U}\mathbf{V}^{\top},$$

then by triangle inequality,

$$\|\hat{\mathbf{M}} - \mathbf{M}\|_\mathrm{F} \le \|\hat{\mathbf{M}} - \mathcal{T}_{r'}(\mathbf{M})\|_\mathrm{F} + \|\mathcal{T}_{r'}(\mathbf{M}) - \mathbf{M}\|_\mathrm{F}.$$

For the first term, on the event $\|\hat{\mathbf{M}} - \mathbf{M}\|_\mathrm{op} \lesssim \sqrt{n_1 p}$, the first term on the right hand side of the previous equation is bounded by

$$
\begin{aligned}
\|\hat{\mathbf{M}} - \mathcal{T}_{r'}(\mathbf{M})\|_\mathrm{F} &\le \sqrt{r'}\|\hat{\mathbf{M}} - \mathcal{T}_{r'}(\mathbf{M})\|_\mathrm{op} \\
&\le \sqrt{n_2 p}(\|\hat{\mathbf{M}} - \mathbf{M}\|_\mathrm{op} + \|\mathbf{M} - \mathcal{T}_{r'}(\mathbf{M})\|_\mathrm{op}) \\
&\lesssim \sqrt{n_2 p}(\sqrt{n_1 p} + \sigma_{r'+1}(\mathbf{M})) \\
&\lesssim \sqrt{n_1 n_2 p^2}.
\end{aligned}
$$

where we have applied (18) in the last inequality. Now for the other term,

$$\|\mathcal{T}_{r'}(\mathbf{M}) - \mathbf{M}\|_\mathrm{F} \le 2\|\mathbf{M}\|_\mathrm{F} \le 2\sqrt{\mathrm{tr}(\mathbf{M}^\top \mathbf{M})} \le 2\sqrt{n_1 n_2 p^2}.$$

Therefore, $\|\hat{\mathbf{M}} - \mathbf{M}\|_\mathrm{F} \lesssim \sqrt{n_1 n_2 p^2}$ with probability at least $1 - n_1^{-1}$.

# B    Proof of Theorem 2

We denote the output of Theorem 1 by $\hat{\mathbf{M}}_1$ and the output of Theorem 2 by $\hat{\mathbf{M}}_2$. We will prove the following result on the event $\|\mathbf{A}_\mathrm{re} - \mathbf{M}\|_\mathrm{op} \lesssim \sqrt{n_1 p}$.

**For operator norm.** To prove $\|\hat{\mathbf{M}}_2 - \mathbf{M}\|_\mathrm{op} \lesssim \sqrt{n_1 p}$, it is sufficient to show that $\|\hat{\mathbf{M}}_1 - \hat{\mathbf{M}}_2\|_\mathrm{op} \lesssim \sqrt{n_1 p}$. Using the definition of these two estimators,

$$\|\hat{\mathbf{M}}_1 - \hat{\mathbf{M}}_2\|_\mathrm{op} = \sigma_{r'+1}(\mathbf{A}_\mathrm{re}).$$

Then the proof is complete by applying (19). Now we need to show $\|\hat{\mathbf{M}}_2 - \mathbf{M}\|_\mathrm{op} \lesssim \sqrt{n_1 n_2 p^2}$. Since the operator norm is bounded by the Frobenius norm, we only need to prove $\|\hat{\mathbf{M}}_2 - \mathbf{M}\|_\mathrm{F} \lesssim \sqrt{n_1 n_2 p^2}$. See the following proof for this bound.

**For Frobenius norm.** Case 1: $r' = r$. Applying (18), we have

$$\|\hat{\mathbf{M}}_1 - \hat{\mathbf{M}}_2\|_\mathrm{F} \le \sigma_{r'+1}r' \lesssim \sqrt{n_1 p r}.$$

Combining the result of Theorem 3, it shows $\|\hat{\mathbf{M}}_2 - \mathbf{M}\|_\mathrm{F} \lesssim \sqrt{n_1 p r}$.

Case 2: $r' = \lfloor n_2 p \rfloor$. Since the inequality $\|\hat{\mathbf{M}}_2 - \mathbf{M}\|_\mathrm{op} \lesssim \sqrt{n_1 p}$ still holds, the proof is identical the Case 2 for Frobenius norm of the proof of Theorem 1.

# C    Proof of Theorem 3

Firstly, we will prove (7). The proof is an application of Fano's inequality. We assume $n_1 \ge n_2$ without loss of generality in this proof. We first derive the packing number of the parameter space $\boldsymbol{\Theta} = \boldsymbol{\Theta}_1(n_1, n_2, p, r)$ equipped with Frobenius norm.

**Lemma 1.** *For $p \in (0, 1]$ and positive integers $n_1, n_2 \ge r$, there exists a finite subset of the parameter space $\boldsymbol{\Theta}_1(n_1, n_2, p, r)$ satisfying*

   (a) *The cardinality of this subset is at least $\exp\left(\frac{n_1 r}{5}\right)$.*

   (b) *For every $\mathbf{M}$ and $\tilde{\mathbf{M}}$ in this subset, $\frac{(n_1 p r) \wedge (n_1 n_2 p^2)}{5000} \le \|\mathbf{M} - \tilde{\mathbf{M}}\|_F^2 \le \frac{n_1 p r}{625}$.*

   (c) *For every $\mathbf{M}$ and $\tilde{\mathbf{M}}$ in this subset, $\mathbf{M}_{ij} = 0$ if and only if $\tilde{\mathbf{M}}_{ij} = 0$. That is, $\{(i, j) : \mathbf{M}_{ij} = 0\} = \{(i, j) : \tilde{\mathbf{M}}_{ij} = 0\}$*

   (d) *For $\mathbf{M}$ in this subset, if $\mathbf{M} \ne 0$, then $\mathbf{M}_{ij} \in \left[\frac{12p}{25}, \frac{13p}{25}\right]$.*

*Proof.* Let us define random matrix

$$\mathbf{M} = \frac{p}{2}(\mathbf{1}_{n_1 \times (r\lfloor \frac{n_2}{r} \wedge \frac{1}{p} \rfloor)}, \mathbf{O}) + \frac{1}{50}p(\mathbf{U}, \dots, \mathbf{U}, \mathbf{O})$$

where $\mathbf{U} \in \mathbb{R}^{n_1 \times r}$ with i.i.d. rademacher entries and $\mathbf{U}$ is repeated $\lfloor \frac{n_2}{r} \wedge \frac{1}{p} \rfloor$ many times, and $\mathbf{O}$ is a zero matrix with dimension $n_1 \times \left(n_2 - r\lfloor \frac{n_2}{r} \wedge \frac{1}{p} \rfloor\right)$. Let $\tilde{\mathbf{U}}$ be an independent copy of $\mathbf{U}$, and construct $\tilde{\mathbf{M}}$ by $\tilde{\mathbf{U}}$ as an independent copy of $\mathbf{M}$. In particular, $\mathbf{M}_{ij} \in \left\{0, \frac{12p}{25}, \frac{13p}{25}\right\}$, so condition (c) and (d) satisfied. Then $\|\mathbf{U} - \tilde{\mathbf{U}}\|_{\mathrm{F}}^2 \leq 4n_1 r$. Therefore,

$$\|\mathbf{M} - \tilde{\mathbf{M}}\|_{\mathrm{F}}^2 = \frac{1}{2500}p^2\lfloor \frac{n_2}{r} \wedge \frac{1}{p} \rfloor\|\mathbf{U} - \tilde{\mathbf{U}}\|_{\mathrm{F}}^2 \leq \frac{n_1 pr}{625}.$$

Hence, the upper bound of condition (b) is satisfied. On the other hand, since $\frac{n_2}{r} \wedge \frac{1}{p} \geq 1$, $\lfloor \frac{n_2}{r} \wedge \frac{1}{p} \rfloor \geq \frac{1}{2}\left(\frac{n_2}{r} \wedge \frac{1}{p}\right)$. Thus,

$$\|\mathbf{M} - \tilde{\mathbf{M}}\|_{\mathrm{F}}^2 = \frac{1}{2500}p^2\lfloor \frac{n_2}{r} \wedge \frac{1}{p} \rfloor\|\mathbf{U} - \tilde{\mathbf{U}}\|_{\mathrm{F}}^2 \geq \frac{1}{5000}\left(p \wedge \frac{n_2 p^2}{r}\right)\|\mathbf{U} - \tilde{\mathbf{U}}\|_{\mathrm{F}}^2.$$

By Hoeffding's inequality,

$$\mathbb{P}\left(\|\mathbf{U} - \tilde{\mathbf{U}}\|_{\mathrm{F}}^2 \leq n_1 r\right) = \mathbb{P}\left(\frac{1}{n_1}\sum_{i=1}^{n_1}\sum_{j=1}^{r}(\varepsilon_{ij} - \tilde{\varepsilon}_{ij})^2 \leq r\right)$$

$$= \mathbb{P}\left(\frac{1}{2}\sum_{i=1}^{n_1}\sum_{j=1}^{r}[(\varepsilon_{ij} - \tilde{\varepsilon}_{ij})^2 - 2] \leq \frac{n_1(r - 2r)}{2}\right)$$

$$\leq \exp\left(-\frac{n_1 r}{2}\right).$$

Suppose $\|\mathbf{U} - \tilde{\mathbf{U}}\|_{\mathrm{F}}^2 > n_1 r$, then $\|\mathbf{M} - \tilde{\mathbf{M}}\|_{\mathrm{F}}^2 > \frac{(n_1 pr) \wedge (n_1 n_2 p^2)}{5000}$ gives the lower bound of condition (b). Now we consider $N = e^{n_1 r/5}$ i.i.d. copies. Let $\mathbf{M}^{(m)}, m \in [N]$ be $N$ independent copies of $\mathbf{M}$, then we have

$$\mathbb{P}\left(\min_{m,m' \in [N]}\|\mathbf{M}^{(m)} - \mathbf{M}^{(m')}\|_F^2 > \frac{(n_1 pr) \wedge (n_1 n_2 p^2)}{5000}\right) \geq 1 - N^2 \exp\left(-\frac{n_1 r}{2}\right)$$

$$\geq 1 - \exp\left(-\frac{n_1 r}{10}\right).$$

Therefore, we can draw $N$ i.i.d. copies of $\mathbf{M}$ to fulfill the requirements in the lemma with positive probability. $\qquad\square$

Now we introduce the Fano's inequality. We will use the version provided by [24] in our proofs.

**Lemma 2** (Fano's inequality). *Assume $N \geq 3$ and suppose $\{\theta_1, \dots, \theta_N\} \subset \Theta$ such that*

  *(i) for all $1 \leq i < j \leq N$, $d(\theta_i, \theta_j) \geq 2\alpha$, where $d$ is a metric on $\Theta$;*

  *(ii) let $P_i$ be the distribution with respect to parameter $\theta_i$, then for all $i, j \in [N]$, $P_i$ is absolutely continuous with respect to $P_j$;*

  *(iii) for all $i, j \in N$, the Kullback-Leibler divergence $D_{KL}(P_i\|P_j) \leq \beta \log(N - 1)$ for some $0 < \beta < 1/8$.*

*Then*

$$\inf_{\hat{\theta}} \sup_{\theta \in \Theta} \mathbb{P}(d(\hat{\theta}, \theta) \geq \alpha) \geq \frac{\sqrt{N - 1}}{1 + \sqrt{N - 1}}\left(1 - 2\beta - \sqrt{\frac{2\beta}{\log(N - 1)}}\right). \tag{20}$$

**Lemma 3.** *For random adjacency matrix model* (1) *with parameters* $\mathbf{M}, \tilde{\mathbf{M}} \in [a,b]^{n_1 \times n_2}$, *their Kullback-Leibler divergence is upper bounded by*

$$D_{KL}(P_{\mathbf{M}} \| P_{\tilde{\mathbf{M}}}) \leq \frac{\|\mathbf{M} - \tilde{\mathbf{M}}\|_F^2}{a(1-b)}.$$

*Proof.* We firstly consider entrywise KL-divergence. For $p, q \in [a,b]$,

$$
\begin{aligned}
D_{KL}(\text{Ber}(p) \| \text{Ber}(q)) &= p \log \frac{p}{q} + (1-p) \log \frac{1-p}{1-q} \\
&= p \log \left( 1 + \frac{p-q}{q} \right) + (1-p) \log \left( 1 - \frac{p-q}{1-q} \right) \\
&\leq p \left( \frac{p-q}{q} \right) + (1-p) \left( -\frac{p-q}{1-q} \right) \\
&= \frac{p(p-q)(1-q) - q(1-p)(p-q)}{q(1-q)} \\
&= \frac{(p-q)^2}{q(1-q)} \leq \frac{(p-q)^2}{a(1-b)}.
\end{aligned}
$$

By independence of each entry, we have $D_{KL}(P_{\mathbf{M}} \| P_{\tilde{\mathbf{M}}}) \leq \frac{\|\mathbf{M} - \tilde{\mathbf{M}}\|_F^2}{a(1-b)}$. $\square$

Now we are ready to prove (7). Let $\Theta$ in Lemma 2 with $N = \exp\left(\frac{n_1 r}{5}\right)$. For distinct $\mathbf{M}, \tilde{\mathbf{M}} \in \Theta$,

$$D_{KL}(P_{\mathbf{M}} \| P_{\tilde{\mathbf{M}}}) \leq \frac{\|\mathbf{M} - \tilde{\mathbf{M}}\|_F^2}{\left(\frac{12}{25} p\right)\left(1 - \frac{13}{25} p\right)} \leq \frac{n_1 p r}{625 \left(\frac{12}{25} p\right)\left(1 - \frac{13}{25} p\right)} \leq \frac{n_1 r}{144}.$$

Let $\beta = 1/24$. For $n_1 \geq 10$, $\log(N-1) \geq n_1 r / 6$. Therefore,

$$D_{KL}(P_{\mathbf{M}} \| P_{\tilde{\mathbf{M}}}) \leq \frac{n_1 r}{144} \leq \beta \log(N-1).$$

On the other hand, the lower bound on the Frobenius norm satisfies

$$\|\mathbf{M} - \tilde{\mathbf{M}}\|_F \geq 2\alpha := \sqrt{\frac{(n_1 p r) \wedge (n_1 n_2 p^2)}{5000}}.$$

and $D_{KL}(P_{\mathbf{M}} \| P_{\tilde{\mathbf{M}}}) \leq \beta n_1 r / 6$ for every pair of distinct elements $\mathbf{M}$ and $\tilde{\mathbf{M}}$ in the subset. Then by (20) and straightforward algebra,

$$\inf_{\hat{\mathbf{M}}} \sup_{i=1,2,\dots N} \mathbb{P} \left( \|\hat{\mathbf{M}} - \mathbf{M}\|_F^2 \geq \frac{(n_1 p r) \wedge (n_1 n_2 p^2)}{20000} \right) \geq \frac{1}{2}.$$

To verify (6), it suffices to observe that

$$\|\hat{\mathbf{M}} - \mathbf{M}\|_F^2 \geq \|\hat{\mathbf{M}} - \mathbf{M}\|_{op}^2.$$

for any $\hat{\mathbf{M}}$ and $\mathbf{M}$ and consider a restriction on the submodel $\Theta = \Theta_1(n_1, n_2, p, 1)$.

# D  Proof of Theorem 4

**Lemma 4** (Davis-Kahan theorem for eigenspaces). *For symmetric matrices* $\mathbf{M}, \hat{\mathbf{M}} \in \mathbb{R}^{n \times n}$, *suppose* $\mathbf{M} = \mathbf{U}_1 \mathbf{\Lambda}_1 \mathbf{U}_1^\top + \mathbf{U}_2 \mathbf{\Lambda}_2 \mathbf{U}_2^\top$ *and* $\hat{\mathbf{M}} = \hat{\mathbf{U}}_1 \hat{\mathbf{\Lambda}}_1 \hat{\mathbf{U}}_1^\top + \hat{\mathbf{U}}_2 \hat{\mathbf{\Lambda}}_2 \hat{\mathbf{U}}_2^\top$ *where* $(\mathbf{U}_1, \mathbf{U}_2), (\hat{\mathbf{U}}_1, \hat{\mathbf{U}}_2) \in \mathbb{R}^{n_1 \times n_2}$ *are orthogonal. Suppose the singular values of* $\mathbf{\Lambda}_1$ *are contained in the interval* $[a, b]$, *and the singular values of* $\hat{\mathbf{\Lambda}}_2$ *are excluded from* $(a - \delta, b + \delta)$, *then*

$$\|\hat{\mathbf{U}}_2^\top \mathbf{U}_1\| \leq \frac{\|\hat{\mathbf{M}} - \mathbf{M}\| + \|\hat{\mathbf{\Lambda}}_2 - \mathbf{\Lambda}_2\|}{\delta} \tag{21}$$

*for* $\| \cdot \|$ *is either Frobenius norm or operator norm.*

*Proof.* Since $\mathbf{U}_1^\top \mathbf{U}_1 = \mathbf{I}$ and $\mathbf{U}_2^\top \mathbf{U}_1 = 0$,

$$\mathbf{M}\mathbf{U}_1 = (\mathbf{U}_1 \boldsymbol{\Lambda}_1 \mathbf{U}_1^\top + \mathbf{U}_2 \boldsymbol{\Lambda}_2 \mathbf{U}_2^\top)\mathbf{U}_1 = \mathbf{U}_1 \boldsymbol{\Lambda}_1.$$

In the same way, we have $\hat{\mathbf{U}}_2^\top \hat{\mathbf{M}} = \hat{\boldsymbol{\Lambda}}_2 \hat{\mathbf{U}}_2^\top$. It follows that

$$\hat{\mathbf{U}}_2^\top (\hat{\mathbf{M}} - \mathbf{M})\mathbf{U}_1 = \hat{\mathbf{U}}_2^\top \hat{\mathbf{M}}\mathbf{U}_1 - \hat{\mathbf{U}}_2^\top \mathbf{M}\mathbf{U}_1^\top = \hat{\boldsymbol{\Lambda}}_2 \hat{\mathbf{U}}_2^\top \mathbf{U}_1 - \hat{\mathbf{U}}_2^\top \mathbf{U}_1 \boldsymbol{\Lambda}_1. \tag{22}$$

Since $\mathbf{U}_1$ and $\hat{\mathbf{U}}_2$ have orthonormal columns,

$$\|(\boldsymbol{\Lambda}_2 - \hat{\boldsymbol{\Lambda}}_2)\hat{\mathbf{U}}_2^\top \mathbf{U}_1\| \le \|\boldsymbol{\Lambda}_2 - \hat{\boldsymbol{\Lambda}}_2\|\|\hat{\mathbf{U}}_2^\top \mathbf{U}_1\|_{\mathrm{op}} \le \|\boldsymbol{\Lambda}_2 - \hat{\boldsymbol{\Lambda}}_2\|. \tag{23}$$

We combine (22) and (23), for any real number $c$,

$$\begin{aligned}
\|\boldsymbol{\Lambda}_2 - \hat{\boldsymbol{\Lambda}}_2\| + \|\hat{\mathbf{U}}_2(\hat{\mathbf{M}} - \mathbf{M})\mathbf{U}_1\| &\ge \|(\boldsymbol{\Lambda}_2 - \hat{\boldsymbol{\Lambda}}_2)\hat{\mathbf{U}}_2^\top \mathbf{U}_1\| + \|\hat{\boldsymbol{\Lambda}}_2 \hat{\mathbf{U}}_2^\top \mathbf{U}_1 - \hat{\mathbf{U}}_2 \mathbf{U}_1 \boldsymbol{\Lambda}_1\| \\
&\ge \|\boldsymbol{\Lambda}_2 \hat{\mathbf{U}}_2^\top \mathbf{U}_1 - \hat{\mathbf{U}}_2^\top \mathbf{U}_1 \boldsymbol{\Lambda}_1\| \\
&= \|(\boldsymbol{\Lambda}_2 - c\mathbf{I})\hat{\mathbf{U}}_2^\top \mathbf{U}_1 - \hat{\mathbf{U}}_2 \mathbf{U}_1 (\boldsymbol{\Lambda}_1 - c\mathbf{I})\| \\
&\ge \|(\boldsymbol{\Lambda}_2 - c\mathbf{I})\hat{\mathbf{U}}_2^\top \mathbf{U}_1\| - \|\hat{\mathbf{U}}_2 \mathbf{U}_1 (\boldsymbol{\Lambda}_1 - c\mathbf{I})\|.
\end{aligned}$$

Now we let $c = (a+b)/2$ and $r = (b-a)/2$, then the eigenvalues of $\boldsymbol{\Lambda}_1 - c\mathbf{I}$ are contained in $[-r, r]$ and the eigenvalues of $\hat{\boldsymbol{\Lambda}}_2 - c\mathbf{I}$ are excluded from $(-r-\delta, r+\delta)$. Therefore,

$$\|(\boldsymbol{\Lambda}_2 - c\mathbf{I})\hat{\mathbf{U}}_2^\top \mathbf{U}_1\| \ge \frac{1}{\|(\boldsymbol{\Lambda}_2 - c\mathbf{I})^{-1}\|_{\mathrm{op}}}\|\hat{\mathbf{U}}_2^\top \mathbf{U}_1\| \ge (r+\delta)\|\hat{\mathbf{U}}_2^\top \mathbf{U}_1\|,$$

and

$$\|\hat{\mathbf{U}}_2 \mathbf{U}_1 (\boldsymbol{\Lambda}_1 - c\mathbf{I})\| \le \|\hat{\mathbf{U}}_2 \mathbf{U}_1\|\|\boldsymbol{\Lambda}_1 - c\mathbf{I}\|_{\mathrm{op}} \le r\|\hat{\mathbf{U}}_2 \mathbf{U}_1\|.$$

Hence, we can conclude that

$$\|\boldsymbol{\Lambda}_2 - \hat{\boldsymbol{\Lambda}}_2\| + \|\hat{\mathbf{U}}_2(\hat{\mathbf{M}} - \mathbf{M})\mathbf{U}_1\| \ge (r+\delta)\|\hat{\mathbf{U}}_2^\top \mathbf{U}_1\| - r\|\hat{\mathbf{U}}_2^\top \mathbf{U}_1\| \ge \delta\|\hat{\mathbf{U}}_2^\top \mathbf{U}_1\|.$$

$\|\hat{\mathbf{U}}_2(\hat{\mathbf{M}} - \mathbf{M})\mathbf{U}_1\| \le \|\hat{\mathbf{U}}_2(\hat{\mathbf{M}} - \mathbf{M})(\mathbf{U}_1, \mathbf{U}_2)\| = \|\hat{\mathbf{U}}_2(\hat{\mathbf{M}} - \mathbf{M})\|$, and similarly, $\|\hat{\mathbf{U}}_2(\hat{\mathbf{M}} - \mathbf{M})\| \le \|\hat{\mathbf{M}} - \mathbf{M}\|$. Hence (21) is obtained. $\qquad\square$

**Corollary 1** (Wedin's Theorem). *For real-valued matrices $\mathbf{M}, \hat{\mathbf{M}} \in \mathbb{R}^{n_1 \times n_2}$, suppose that $\mathbf{M} = \mathbf{U}_1 \boldsymbol{\Lambda}_1 \mathbf{V}_1^\top + \mathbf{U}_2 \boldsymbol{\Lambda}_2 \mathbf{V}_2^\top$ and $\hat{\mathbf{M}} = \hat{\mathbf{U}}_1 \hat{\boldsymbol{\Lambda}}_1 \hat{\mathbf{V}}_1^\top + \hat{\mathbf{U}}_2 \hat{\boldsymbol{\Lambda}}_2 \hat{\mathbf{V}}_2^\top$ are the singular value decompositions so that $(\mathbf{U}_1, \mathbf{U}_2), (\hat{\mathbf{U}}_1, \hat{\mathbf{U}}_2) \in \mathbb{R}^{n_1 \times n_1}, (\mathbf{V}_1, \mathbf{V}_2), (\hat{\mathbf{V}}_1, \hat{\mathbf{V}}_2) \in \mathbb{R}^{n_2 \times n_2}$ are orthogonal, and $\boldsymbol{\Lambda}_1, \boldsymbol{\Lambda}_2$ are diagonal. Suppose*

$$0 \le \min(diag(\boldsymbol{\Lambda}_1)) \le \max(diag(\boldsymbol{\Lambda}_1)) \le a < a + \delta \le \min(diag(\boldsymbol{\Lambda}_2))$$

*and $\boldsymbol{\Lambda}_2$ and $\hat{\boldsymbol{\Lambda}}_2$ contain top-$r$ singular values of $\mathbf{M}$ of $\hat{\mathbf{M}}$ respectively, then*

$$\max(\|\mathbf{U}_2 \mathbf{U}_2^\top - \hat{\mathbf{U}}_2 \hat{\mathbf{U}}_2^\top\|, \|\mathbf{V}_2 \mathbf{V}_2^\top - \hat{\mathbf{V}}_2 \hat{\mathbf{V}}_2^\top\|) \le \frac{2\|\hat{\mathbf{M}} - \mathbf{M}\|}{\delta} \tag{24}$$

*for $\|\cdot\|$ is either Frobenius norm or operator norm.*

*Proof.* We consider the symmetric dilation of $\mathbf{M}$, given by

$$\mathbf{M}^\dagger = \begin{pmatrix} 0 & \mathbf{M} \\ \mathbf{M}^\top & 0 \end{pmatrix}. \tag{25}$$

By Lemma 2(a) of [28], we let

$$\mathbf{W}_1 = \begin{pmatrix} \mathbf{U}_1 & \mathbf{U}_1 \\ \mathbf{V}_1 & -\mathbf{V}_1 \end{pmatrix}, \quad \mathbf{W}_2 = \begin{pmatrix} \mathbf{U}_2 & \mathbf{U}_2 \\ \mathbf{V}_2 & -\mathbf{V}_2 \end{pmatrix}, \quad \boldsymbol{\Sigma}_1 = \begin{pmatrix} \boldsymbol{\Lambda}_1 & 0 \\ 0 & -\boldsymbol{\Lambda}_1 \end{pmatrix}, \quad \boldsymbol{\Sigma}_2 = \begin{pmatrix} \boldsymbol{\Lambda}_2 & 0 \\ 0 & -\boldsymbol{\Lambda}_2, \end{pmatrix}$$

then we have the decomposition

$$\mathbf{M}^\dagger = \frac{1}{2}[\mathbf{W}_1 \boldsymbol{\Sigma}_1 \mathbf{W}^\top + \mathbf{W}_2 \boldsymbol{\Sigma}_2 \mathbf{W}_2],$$

and similarly,

$$\hat{\mathbf{M}}^\dagger = \frac{1}{2}[\hat{\mathbf{W}}_1 \hat{\boldsymbol{\Sigma}}_1 \hat{\mathbf{W}}^\top + \hat{\mathbf{W}}_2 \hat{\boldsymbol{\Sigma}}_2 \hat{\mathbf{W}}_2],$$

where

$$\hat{\mathbf{W}}_1 = \begin{pmatrix} \hat{\mathbf{U}}_1 & \hat{\mathbf{U}}_1 \\ \hat{\mathbf{V}}_1 & -\hat{\mathbf{V}}_1 \end{pmatrix}, \quad \hat{\mathbf{W}}_2 = \begin{pmatrix} \hat{\mathbf{U}}_2 & \hat{\mathbf{U}}_2 \\ \hat{\mathbf{V}}_2 & -\hat{\mathbf{V}}_2 \end{pmatrix}, \quad \hat{\boldsymbol{\Sigma}}_1 = \begin{pmatrix} \hat{\boldsymbol{\Lambda}}_1 & 0 \\ 0 & -\hat{\boldsymbol{\Lambda}}_1 \end{pmatrix}, \quad \hat{\boldsymbol{\Sigma}}_2 = \begin{pmatrix} \hat{\boldsymbol{\Lambda}}_2 & 0 \\ 0 & -\hat{\boldsymbol{\Lambda}}_2 \end{pmatrix},$$

It is easy to check that $\|\hat{\mathbf{M}}^\dagger - \mathbf{M}^\dagger\|_{\text{op}} \leq \|\hat{\mathbf{M}} - \mathbf{M}\|_{\text{op}}$ and $\|\hat{\boldsymbol{\Sigma}}_2 - \boldsymbol{\Sigma}_2\|_{\text{op}} \leq \|\hat{\boldsymbol{\Lambda}} - \boldsymbol{\Lambda}\|_{\text{op}}$. Since $\boldsymbol{\Lambda}_2$ has eigenvalues contained in $[0, a]$, the eigenvalues of $\boldsymbol{\Sigma}_2$ are contained in $[-a, a]$. By Lemma 4,

$$\|\mathbf{W}_1^\top \mathbf{W}_2\|_{\text{op}} \leq \frac{\|\hat{\mathbf{M}}^\dagger - \mathbf{M}^\dagger\|_{\text{op}} + \|\hat{\boldsymbol{\Sigma}}_2 - \boldsymbol{\Sigma}_2\|_{\text{op}}}{\delta} = \frac{\|\hat{\mathbf{M}} - \mathbf{M}\|_{\text{op}} + \|\hat{\boldsymbol{\Lambda}}_2 - \boldsymbol{\Lambda}_2\|_{\text{op}}}{\delta}.$$

By Lemma 1 of [4],

$$\begin{aligned}
\|\mathbf{W}_1^\top \mathbf{W}_2\|_{\text{op}} &\geq \frac{1}{2} \|\mathbf{W}_2 \mathbf{W}_2^\top - \hat{\mathbf{W}}_2 \hat{\mathbf{W}}_2^\top\|_{\text{op}} \\
&= \left\| \begin{pmatrix} \mathbf{U}_2 \mathbf{U}_2^\top - \hat{\mathbf{U}}_2 \hat{\mathbf{U}}_2^\top & 0 \\ 0 & \mathbf{V}_2 \mathbf{V}_2^\top - \hat{\mathbf{V}}_2 \hat{\mathbf{V}}_2^\top \end{pmatrix} \right\|_{\text{op}} \\
&= \max(\|\mathbf{U}_2 \mathbf{U}_2^\top - \hat{\mathbf{U}}_2 \hat{\mathbf{U}}_2^\top\|_{\text{op}}, \|\mathbf{V}_2 \mathbf{V}_2^\top - \hat{\mathbf{V}}_2 \hat{\mathbf{V}}_2^\top\|_{\text{op}}).
\end{aligned}$$

Hence we obtain

$$\max(\|\mathbf{U}_2 \mathbf{U}_2^\top - \hat{\mathbf{U}}_2 \hat{\mathbf{U}}_2^\top\|_{\text{op}}, \|\mathbf{V}_2 \mathbf{V}_2^\top - \hat{\mathbf{V}}_2 \hat{\mathbf{V}}_2^\top\|_{\text{op}}) \leq \frac{\|\hat{\mathbf{M}} - \mathbf{M}\|_{\text{op}} + \|\hat{\boldsymbol{\Lambda}}_2 - \boldsymbol{\Lambda}_2\|_{\text{op}}}{\delta}.$$

By Corollary 2, the right hand side is upper bounded by $2\|\hat{\mathbf{M}} - \mathbf{M}\|_{\text{op}}/\delta$. This proves (24) for operator norm. For Frobenius norm, we have $\|\hat{\mathbf{M}}^\dagger - \mathbf{M}^\dagger\|_{\text{F}} \leq \sqrt{2}\|\hat{\mathbf{M}} - \mathbf{M}\|_{\text{F}}$ and $\|\hat{\boldsymbol{\Sigma}}_2 - \boldsymbol{\Sigma}_2\|_{\text{F}} \leq \sqrt{2}\|\hat{\boldsymbol{\Lambda}} - \boldsymbol{\Lambda}\|_{\text{F}}$. By Lemma 4,

$$\|\mathbf{W}_1^\top \mathbf{W}_2\|_{\text{F}} \leq \frac{\|\hat{\mathbf{M}}^\dagger - \mathbf{M}^\dagger\|_{\text{F}} + \|\hat{\boldsymbol{\Sigma}}_2 - \boldsymbol{\Sigma}_2\|_{\text{F}}}{\delta} = \frac{\sqrt{2}\|\hat{\mathbf{M}} - \mathbf{M}\|_{\text{F}} + \sqrt{2}\|\hat{\boldsymbol{\Lambda}}_2 - \boldsymbol{\Lambda}_2\|_{\text{F}}}{\delta}.$$

By Wielandt-Hoffman Theorem [22], $\|\hat{\boldsymbol{\Lambda}}_2 - \boldsymbol{\Lambda}_2\|_{\text{F}} \leq \|\hat{\mathbf{M}} - \mathbf{M}\|_{\text{F}}$. Therefore, the right hand side is upper bounded by $2\sqrt{2}\|\hat{\mathbf{M}} - \mathbf{M}\|_{\text{F}}/\delta$. By Lemma 1 of [4] again,

$$\begin{aligned}
\|\mathbf{W}_1^\top \mathbf{W}_2\|_{\text{F}} &= \frac{1}{\sqrt{2}} \|\mathbf{W}_2 \mathbf{W}_2^\top - \hat{\mathbf{W}}_2 \hat{\mathbf{W}}_2^\top\|_{\text{F}} \\
&= \sqrt{2} \left\| \begin{pmatrix} \mathbf{U}_2 \mathbf{U}_2^\top - \hat{\mathbf{U}}_2 \hat{\mathbf{U}}_2^\top & 0 \\ 0 & \mathbf{V}_2 \mathbf{V}_2^\top - \hat{\mathbf{V}}_2 \hat{\mathbf{V}}_2^\top \end{pmatrix} \right\|_{\text{F}} \\
&= \sqrt{2\|\mathbf{U}_2 \mathbf{U}_2^\top - \hat{\mathbf{U}}_2 \hat{\mathbf{U}}_2^\top\|_{\text{F}}^2 + 2\|\mathbf{V}_2 \mathbf{V}_2^\top - \hat{\mathbf{V}}_2 \hat{\mathbf{V}}_2^\top\|_{\text{F}}^2} \\
&\geq \sqrt{2} \max(\|\mathbf{U}_2 \mathbf{U}_2^\top - \hat{\mathbf{U}}_2 \hat{\mathbf{U}}_2^\top\|_{\text{F}}, \|\mathbf{V}_2 \mathbf{V}_2^\top - \hat{\mathbf{V}}_2 \hat{\mathbf{V}}_2^\top\|_{\text{F}}).
\end{aligned}$$

This completes the proof of (24). $\qquad\square$

**Theorem 6** (Weyl's inequality, Corollary III.2.6 of [1]). *Suppose* $\mathbf{A}$ *and* $\mathbf{B}$ *are* $n \times n$ *real symmetric matrices and let* $\sigma_1(A) \geq \sigma_2(A) \geq \ldots, \geq \sigma_n(\mathbf{A})$ *and* $\sigma_1(\mathbf{B}) \geq \sigma_2(\mathbf{B}) \geq \ldots, \geq \sigma_n(\mathbf{B})$ *be the eigenvalues of* $\mathbf{A}$ *and* $\mathbf{B}$ *respectively, then*

$$\max_{i=1,\ldots,n} |\sigma_i(\mathbf{A}) - \sigma_i(\mathbf{B})| \leq \|\mathbf{A} - \mathbf{B}\|_{op}. \tag{26}$$

**Corollary 2.** *Suppose* $\mathbf{A}$ *and* $\mathbf{B}$ *are not necessarily symmetric and* $\sigma_i(\mathbf{A})$ *and* $\sigma_i(\mathbf{B})$ *are singular values, the inequality* (26) *still holds.*

*Proof.* We consider the symmetric dilation (25) of $\mathbf{A}$ and $\mathbf{B}$, denoted by $\mathbf{A}^\dagger$ and $\mathbf{B}^\dagger$ respectively. Then $\mathbf{A}^\dagger$ has eigenvalues $\sigma_1(\mathbf{A}) \geq \sigma_2(\mathbf{A}) \geq \cdots \geq \sigma_n(\mathbf{A}) \geq 0 \geq -\sigma_n(\mathbf{A}) \geq \cdots \geq -\sigma_2(\mathbf{A}) \geq -\sigma_1(\mathbf{A})$. The eigenvalues of $\mathbf{B}^\dagger$ are similar. Then we apply the fact that $\|\mathbf{A} - \mathbf{B}\|_{op} = \|\mathbf{A}^\dagger - \mathbf{B}^\dagger\|_{op}$ and Weyl's inequality to obtain the result. $\qquad\square$

Now we are ready to prove Theorem 4. Let $\mathbf{M} = \mathbf{U}_1\mathbf{\Lambda}_1\mathbf{V}_1^\top + \mathbf{U}_2\mathbf{\Lambda}_2\mathbf{V}_2^\top$ and $\hat{\mathbf{M}} = \hat{\mathbf{U}}_1\hat{\mathbf{\Lambda}}_1\hat{\mathbf{V}}_1^\top + \hat{\mathbf{U}}_2\hat{\mathbf{\Lambda}}_2\hat{\mathbf{V}}_2^\top$ be singular value decompositions of $\mathbf{M}$ and $\hat{\mathbf{M}}$ respectively, where $\mathrm{diag}(\mathbf{\Lambda}_2) = (\sigma_1(\mathbf{M}), \ldots, \sigma_r(\mathbf{M}))$ and $\mathrm{diag}(\hat{\mathbf{\Lambda}}_2) = (\sigma_1(\hat{\mathbf{M}}), \ldots, \sigma_r(\hat{\mathbf{M}}))$ contains top-$r$ singular values. By Corollary 2, we have

$$\|\hat{\mathbf{\Lambda}}_2 - \mathbf{\Lambda}_2\|_{\mathrm{op}} \le \|\hat{\mathbf{M}} - \mathbf{M}\|_{\mathrm{op}}.$$

By Corollary 1,

$$\max(\|\mathbf{U}_2\mathbf{U}_2^\top - \hat{\mathbf{U}}_2\hat{\mathbf{U}}_2^\top\|_{\mathrm{op}}, \|\mathbf{V}_2\mathbf{V}_2^\top - \hat{\mathbf{V}}_2\hat{\mathbf{V}}_2^\top\|_{\mathrm{op}}) \le \frac{\|\hat{\mathbf{M}} - \mathbf{M}\|_{\mathrm{op}} + \|\hat{\mathbf{\Lambda}}_2 - \mathbf{\Lambda}_2\|_{\mathrm{op}}}{\sigma}$$
$$\le \frac{2\|\hat{\mathbf{M}} - \mathbf{M}\|_{\mathrm{op}}}{\sigma}.$$

Now we apply Theorem 2.1 of [16],

$$\mathbb{P}(\|\hat{\mathbf{M}} - \mathbf{M}\|_{\mathrm{op}} \lesssim \sqrt{n_1 p}) \ge 1 - n^{-1}.$$

On the event of $\|\hat{\mathbf{M}} - \mathbf{M}\|_{\mathrm{op}} \lesssim \sqrt{n_1 p}$, we have

$$\max(\|\mathbf{U}_2\mathbf{U}_2^\top - \hat{\mathbf{U}}_2\hat{\mathbf{U}}_2^\top\|_{\mathrm{op}}, \|\mathbf{V}_2\mathbf{V}_2^\top - \hat{\mathbf{V}}_2\hat{\mathbf{V}}_2^\top\|_{\mathrm{op}}) \le \frac{2\|\hat{\mathbf{M}} - \mathbf{M}\|_{\mathrm{op}}}{\sigma}$$
$$\lesssim \frac{\sqrt{n_1 p}}{\sigma}.$$

For Frobenius norm, we have that $\|\hat{\mathbf{\Lambda}}_2 - \mathbf{\Lambda}_2\|_{\mathrm{F}} \le \sqrt{r}\|\hat{\mathbf{\Lambda}}_2 - \mathbf{\Lambda}_2\|_{\mathrm{op}}$ $\le \sqrt{r}\|\hat{\mathbf{M}} - \mathbf{M}\|_{\mathrm{op}}$ by Corollary 2,

$$\max(\|\mathbf{U}_2\mathbf{U}_2^\top - \hat{\mathbf{U}}_2\hat{\mathbf{U}}_2^\top\|_{\mathrm{F}}, \|\mathbf{V}_2\mathbf{V}_2^\top - \hat{\mathbf{V}}_2\hat{\mathbf{V}}_2^\top\|_{\mathrm{F}}) \le \frac{\|\hat{\mathbf{M}} - \mathbf{M}\|_{\mathrm{F}} + \|\hat{\mathbf{\Lambda}}_2 - \mathbf{\Lambda}\|_{\mathrm{F}}}{\sigma}$$
$$\le \frac{2\sqrt{r}\|\hat{\mathbf{M}} - \mathbf{M}\|_{\mathrm{op}}}{\sigma}$$
$$\lesssim \frac{\sqrt{n_1 p r}}{\sigma}.$$

## E  Proof of Theorem 5

We firstly consider the case $r > 1$. Let integer $k_2 \ge 1$, $\sigma > 0$ and $\mu \in (0,1)$ be given by

$$k_2 = \lceil (10/p)^2 \sigma_*^2/n_1 \rceil, \ \sigma^2 = n_1 k_2 (p/10)^2, \ \mu^2 = \min\{21/(2k_2 p), 0.1\}/2. \tag{27}$$

Clearly $\sigma_* \le \sigma \le \sqrt{2}\sigma_*$. As $r\sigma_*^2 \le n_1 n_2 p^2/C_0$, $k_2 \le 200 n_2/(rC_0) \le (n_2 - 1)/(2r - 2)$ for sufficiently large $C_0$. This allows the following construction. Let $\mathbf{H} \in [-\sqrt{3}, \sqrt{3}]^{n_1 \times (r-1)}$ such that $(\mathbf{H}, \mathbf{1}_{n_1})^\top(\mathbf{H}, \mathbf{1}_{n_1})/n_1 = \mathbf{I}_r$. Let $\mathbf{U}_i, i = 1, \ldots, N$, be distinct matrices in $\{-1, 1\}^{n_1 \times (r-1)}$, with $N = 2^{n_1(r-1)}$, $\mathbf{W}_i = \sqrt{1 - \mu^2}\mathbf{H} + \mu\mathbf{U}_i$ with $0 < \mu < 1$, and

$$\mathbf{M}_i = \frac{p}{2}\mathbf{1}_{n_1 \times n_2} + \frac{p}{10}(\mathbf{W}_i, -\mathbf{W}_i, \ldots, \mathbf{W}_i, -\mathbf{W}_i, \mathbf{O}), \tag{28}$$

where $(\mathbf{W}_i, -\mathbf{W}_i)$ is repeated $k_2$ times. As $\|\mathbf{W}_i\|_\infty \le \sqrt{(1 - \mu^2)3} + \mu \le 2$, $\mathbf{M}_i \in [0.3p, 0.7p]^{n_1 \times n_2}$. Let $\mathbf{P}_i = \mathbf{P}_{\mathbf{M}_i} = \mathbf{M}_i\mathbf{M}_i^\dagger \in \mathbb{R}^{n_1 \times n_1}$ be the orthogonal projection to the column space of $\mathbf{M}_i$ and $\mathbf{X}_i = (\mathbf{W}_i, \mathbf{1}_{n_1}) = (\sqrt{1 - \mu^2}\mathbf{H} + \mu\mathbf{U}_i, \mathbf{1}_{n_1}) \in \mathbb{R}^{n_1 \times r}$. When $\mathrm{rank}(\mathbf{X}_i) = r$, $\mathbf{X}_i$ has the same column space as $\mathbf{M}_i$ and $\mathbf{P}_i = \mathbf{X}_i(\mathbf{X}_i^\top\mathbf{X}_i)^{-1}\mathbf{X}_i^\top$. Let

$$\mathbf{V}_{i,j} = \left(\begin{array}{c|c} \mathbf{U}_i^\top\mathbf{U}_j/n_1 & \mathbf{0} \\ \hline \mathbf{0}^\top & 1 \end{array}\right), \quad \mathbf{\Delta}_i = \frac{\mu}{n_1}\left(\begin{array}{c|c} \sqrt{1 - \mu^2}\mathbf{U}_i^\top\mathbf{H} & \mathbf{U}_i^\top\mathbf{1}_{n_1} \\ \hline \mathbf{0}^\top & 0 \end{array}\right),$$

and $\mathbf{\Delta}_{i,j} = \mathbf{\Delta}_i + \mathbf{\Delta}_j^\top + \mu^2(\mathbf{V}_{i,i} - \mathbf{I}_r)I_{\{i=j\}}$. By algebra, we have

$$n_1^{-1}\mathbf{X}_i^\top\mathbf{X}_j = \begin{cases} \mathbf{I}_r + \mathbf{\Delta}_{i,j} + \mu^2(\mathbf{V}_{i,j} - \mathbf{I}_r), & i \ne j, \\ \mathbf{I}_r + \mathbf{\Delta}_{i,i}, & i = j. \end{cases} \tag{29}$$

Thus, $\mathrm{rank}(\mathbf{X}_i) = r$ when $\|\mathbf{\Delta}_{i,i}\|_{\mathrm{op}} < 1$. Let $\sigma_r(\cdot)$ denote the $r$-th largest singular value. We have

$$\sigma_r(\mathbf{M}_i) \geq (p/10)\sqrt{n_1 2k_2(1 - 2\mu^2\Delta_i' - \mu^2\Delta_i'' - (1 + 1/92)\mu^4(\Delta_i''')^2)_+}$$

by Lemma 6, where $\Delta_i' = \|\mathbf{U}_i^\top \mathbf{H}/(n_1\mu)\|_{\mathrm{op}}$, $\Delta_i'' = \|\mathbf{U}_i^\top \mathbf{U}_i/n_1 - \mathbf{I}_{r-1}\|_{\mathrm{op}}$, and $\Delta_i''' = \|\mathbf{U}_i^\top \mathbf{1}_{n_1}/(n_1\mu)\|_2$.

Let $\varepsilon_n$ satisfying $0 < \varepsilon_n \leq 1/(8\mu^2)$ to be determined later and $\Omega^* = \{i \leq N : \Delta_i' \vee \Delta_i'' \vee \Delta_i''' \leq \varepsilon_n\}$. As $\|\mathbf{\Delta}_i\|_{\mathrm{op}} \leq \mu^2\Delta_i' + \mu^2\Delta_i'''$ and $\|\mathbf{V}_{i,i} - \mathbf{I}_r\|_{\mathrm{op}} = \Delta_i''$, we have $\|\mathbf{\Delta}_{i,i}\|_{\mathrm{op}} \leq 5\mu^2\varepsilon_n$ for $i \in \Omega^*$ and

$$\{i \in \Omega^*\} \implies \{\mathbf{M}_i \in [0.3p, 0.7p]^{n_1 \times n_2}, \sigma_r(\mathbf{M}_i) \geq \sigma \geq \sigma_*, \mathrm{rank}(\mathbf{X}_i) = r\}. \tag{30}$$

as $\sigma_r^2(\mathbf{M}_i) \geq (p/10)^2 n_1 2k_2(1 - 4\mu^2\varepsilon_n)_+ \geq (p/10)^2 n_1 k_2 = \sigma^2 \geq \sigma_*^2$ by (27) for $i \in \Omega^*$.

Moreover, for $\{i,j\}$ in $\Omega^*$, $\|\mathbf{\Delta}_{i,j}\|_{\mathrm{op}} \leq (4 + I_{\{i=j\}})\mu^2\varepsilon_n$, so that inserting (29) into $\mathrm{tr}(\mathbf{P}_i\mathbf{P}_j) = \mathrm{tr}((\mathbf{X}_i^\top\mathbf{X}_i)^{-1}\mathbf{X}_i^\top\mathbf{X}_j(\mathbf{X}_j^\top\mathbf{X}_j)^{-1}\mathbf{X}_j^\top\mathbf{X}_i)$ yields

$$\begin{aligned}
\mathrm{tr}(\mathbf{P}_i\mathbf{P}_j) &\leq r + (C_1 - 1)\mu^2\varepsilon_n r + \mu^2(1-\mu^2)\mathrm{tr}(\mathbf{V}_{i,j} + \mathbf{V}_{j,i} - 2\mathbf{I}_r) + \mu^4\mathrm{tr}(\mathbf{V}_{i,j}\mathbf{V}_{j,i} - \mathbf{I}_r) \\
&\leq r + C_1\mu^2\varepsilon_n r + \mu^2(1-\mu^2)\mathrm{tr}(\mathbf{V}_{i,j} + \mathbf{V}_{j,i} - \mathbf{V}_{i,i} - \mathbf{V}_{j,j}) \\
&= r + C_1\mu^2\varepsilon_n r - \mu^2(1-\mu^2)\|\mathbf{U}_i - \mathbf{U}_j\|_{\mathrm{F}}^2/n_1, \qquad \forall\, i,j \in \Omega^*,
\end{aligned} \tag{31}$$

where $C_1$ is a numerical constant. We provide the details of this calculation in Lemma 7.

Let $\mathbf{U}, \mathbf{M}, \mathbf{P}$ be random matrices with the uniform prior distribution $\pi(\cdot)$,

$$\pi(i) = \mathbb{P}_\pi(\mathbf{U} = \mathbf{U}_i, \mathbf{M} = \mathbf{M}_i, \mathbf{P_M} = \mathbf{P}_i) = 1/N = 2^{-n_1(r-1)},$$

so that the elements of $\mathbf{U}$ are i.i.d. Rademacher variables under $\mathbb{P}_\pi$. Let $\mathcal{U}^* = \{\mathbf{U}_i : i \in \Omega^*\}$, $\pi^*$ be the uniform prior on $\Omega^*$ and $\mathbb{P}_{\pi^*}$ the corresponding joint probability so that $\mathbb{P}_{\pi^*}$ is the conditional probability given $\mathbf{U} \in \mathcal{U}^*$ under $\mathbb{P}_\pi$. By (30), $\mathbb{P}_{\pi^*}\{\mathbf{U} \in \mathbf{\Theta}_2(n_1, n_2, p, r, \sigma)\} = 1$ and (12) holds.

It remains to prove (14). By (31) and the details given in Lemma 8, the Frobenius risk of the Bayes estimator under $\mathbb{P}_{\pi^*}$ is bounded by

$$R_{\pi^*}^{\mathrm{Bayes}} = \mathbb{E}_{\pi^*}\left[\|\hat{\mathbf{P}}^* - \mathbf{P_M}\|_{\mathrm{F}}^2\right] \geq \mu^2(1-\mu^2)n_1^{-1}\mathbb{E}_{\pi^*}\left[\|\hat{\mathbf{U}}^* - \mathbf{U}\|_{\mathrm{F}}^2\right] - C_1\mu^2\varepsilon_n r \tag{32}$$

where $\hat{\mathbf{P}}^*$ and $\hat{\mathbf{U}}^*$ are respectively the posterior mean of $\mathbf{P_M}$ and $\mathbf{U}$ under $\mathbb{P}_{\pi^*}$. Moreover, $\|\hat{\mathbf{U}}^*\|_{\mathrm{F}}^2 \vee \|\mathbf{U}\|_{\mathrm{F}}^2 \leq rn_1$ always holds, so that

$$\mathbb{E}_{\pi^*}\left[\|\hat{\mathbf{U}}^* - \mathbf{U}\|_{\mathrm{F}}^2\right] + \mathbb{P}_\pi(\Omega^{*c})4n_1 r \geq \mathbb{E}_\pi\left[\|\hat{\mathbf{U}}^* - \mathbf{U}\|_{\mathrm{F}}^2\right] \geq \mathbb{E}_\pi\left[\|\hat{\mathbf{U}} - \mathbf{U}\|_{\mathrm{F}}^2\right], \tag{33}$$

where $\hat{\mathbf{U}}$ is the Bayes estimator of $\mathbf{U}$ under $\mathbb{P}_\pi$, due to the optimality of $\hat{\mathbf{U}}$ under $\mathbb{P}_\pi$.

Under $\mathbb{P}_\pi$, the elements of $\mathbf{A}$ are independent conditionally on $\mathbf{U}$ and the elements of $\mathbf{U}$ are i.i.d. Rademacher. Moreover, as $(\mathbf{W}_i, -\mathbf{W}_i)$ is repeated $k_2$ times, conditionally on $\mathbf{U}$ the $k_2$ i.i.d. copies of $(A_{i,j}, A_{i,j+r-1})$ are sufficient statistics for the estimation of the $(i,j)$ element $U_{i,j}$ of $\mathbf{U}$ such that $A_{i,j}$ and $A_{i,j+r-1}$ are independent Bernoulli variables with probabilities $p_{i,j} + (\mu p/10)U_{i,j} \in [0.3p, 0.7p]$ and $q_{i,j} - (\mu p/10)U_{i,j} \in [0.3p, 0.7p]$ respectively for some $p_{i,j}$ and $q_{i,j}$ satisfying the constraints. Thus, by Lemma 9, the risk of the Bayes estimator is bounded by

$$\mathbb{E}_\pi\left[(\hat{U}_{i,j} - U_{i,j})^2\right] \geq 1 - 2k_2(\mu p/10)^2/(0.3p(1 - 0.3p)) \geq 1 - 2\mu^2 k_2 p/21.$$

By (27) $\mu^2 = \{(21/(2k_2 p)) \wedge 0.1\}/2$, so that $(1 - \mu^2 2k_2 p/21) \geq 1/2$ and $1 - \mu^2 \geq 0.95$. Thus, by (32) and (33), it follows that

$$\begin{aligned}
R_{\pi^*}^{\mathrm{Bayes}} &\geq \mu^2(1-\mu^2)\left(n_1^{-1}\mathbb{E}_\pi\left[\|\hat{\mathbf{U}} - \mathbf{U}\|_{\mathrm{F}}^2\right] - \mathbb{P}_\pi(\Omega^{*c})4r\right) - C_1\mu^2 r\varepsilon_n \\
&\geq 0.475\mu^2 r - \left(4\mathbb{P}_\pi(\Omega^{*c}) + C_1\varepsilon_n\right)\mu^2 r.
\end{aligned}$$

This gives (14) when $4\mathbb{P}_\pi(\Omega^{*c}) + C_1\varepsilon_n \leq 0.075 = 3/40$. To this end, we pick

$$\varepsilon_n = \max\left\{\sqrt{40\pi r\sigma^2/(n_1^2 p)} + \sqrt{160x_0\sigma^2/(n_1^2 p)}, 4\sqrt{(3r + x_0)/n_1}\right\}$$

with $x_0 = \log(320)$ satisfying $16e^{-x_0} = 0.05$ As $\sigma^2 \leq 2\sigma_*^2 \leq 2n_1 n_2 p^2 r^{-1}/C_0$ and $C_0 r \leq n_1$.

$$\varepsilon_n \leq \max\left\{\sqrt{80\pi p/C_0} + \sqrt{320x_0 p/C_0}, 4\sqrt{(3 + x_0)/C_0}\right\}.$$

Thus, $\mu^2 \varepsilon_n \le 1/8$ and $C_1 \varepsilon_n \le 1/40$ for sufficiently large $C_0$. Moreover, Lemma 5 provides

$$4\mathbb{P}_\pi\{\Omega^{*c}\} \le 16e^{-x_0} \le 1/20,$$

so that $4\mathbb{P}_\pi(\Omega^{*c}) + C_1 \varepsilon_n \le 3/40$ indeed holds. Consequently, by (27)

$$R_{\pi^*}^{\mathrm{Bayes}} \ge 0.4r\mu^2 = 0.2 \min\{21/(2k_2 p), 0.1\} = 0.2 \min\{0.105 n_1 p/\sigma^2, 0.1\}.$$

This gives (14) and completes the proof for $r > 1$.

The proof for $r = 1$ is simpler but the construction is slightly different. Let $\mathbf{u}_i \in \{-1,1\}^{n_1}$, $\mathbf{w}_i = (p/2)\mathbf{1}_{n_1} + (p/10)\mathbf{u}_i$, and $\mathbf{M}_i = \mathbf{w}_i \mathbf{1}_{n_2}^\top$. For $1/C_0 \le 0.16$, we have

$$\mathbf{M}_i \in [0.4p, 0.6p]^{n_1 \times n_2}, \quad \sigma_1^2(\mathbf{M}_i) \ge (0.4p)^2 n_1 n_2 \ge \sigma_*^2, \quad \mathrm{rank}(\mathbf{M}_i) = 1.$$

Let $\mathbf{P}_{\mathbf{M}_i} = \mathbf{w}_i \mathbf{w}_i^\top / \|\mathbf{w}_i\|_2^2$ and $T_{i,j} = \mathbf{w}_i^\top \mathbf{w}_j / n_1$. We have

$$\|\mathbf{P}_{\mathbf{M}_i} - \mathbf{P}_{\mathbf{M}_j}\|_\mathrm{F}^2 = 2(T_{i,i} T_{j,j} - T_{i,j}^2)/T_{i,i} T_{j,j}.$$

Let $\Omega^* = \{i : |\mathbf{u}_i^\top \mathbf{1}_{n_1}/(\mu n_1)| \le \varepsilon_n\}$. For $\{i,j\} \subset \Omega^*$,

$$\begin{aligned}
T_{i,j} &= n_1^{-1}(\mu \mathbf{u}_i + \sqrt{1-\mu^2}\mathbf{1}_{n_1})^\top (\mu \mathbf{u}_j + \sqrt{1-\mu^2}\mathbf{1}_{n_1}) \\
&= n_1^{-1}\big(-\mu^2 \|\mathbf{u}_i - \mathbf{u}_j\|_2^2 + \mu\sqrt{1-\mu^2}(\mathbf{u}_i + \mathbf{u}_j)^\top \mathbf{1}_{n_1}\big) + 1,
\end{aligned}$$

so that $|T_{i,i} - 1| \le 2\mu^2 \varepsilon_n$.

$$\begin{aligned}
&T_{i,i} T_{j,j} - T_{i,j}^2 \\
&= 2n_1^{-1} \mu^2 \|\mathbf{u}_i - \mathbf{u}_j\|_2^2 - n_1^{-2}\mu^4 \|\mathbf{u}_i - \mathbf{u}_j\|_2^4 - \mu^2(1-\mu^2)((\mathbf{u}_i - \mathbf{u}_j)^\top \mathbf{1}_{n_1}/n_1)^2 \\
&\quad + n_1^{-2}\mu^2 \|\mathbf{u}_i - \mathbf{u}_j\|_2^2 \mu\sqrt{1-\mu^2}(\mathbf{u}_i + \mathbf{u}_j)^\top \mathbf{1}_{n_1} \\
&\ge n_1^{-1}\mu^2 \|\mathbf{u}_i - \mathbf{u}_j\|_2^2 (2 - 4\mu^2 - 2\mu^2 \varepsilon_n) - 4\mu^4(1-\mu^2)\varepsilon_n^2.
\end{aligned}$$

We omit the rest of the proof as they are almost identical to the case of $r > 1$.

**Lemma 5.** *Let $\mathbf{H} \in \{-1,1\}^{n_1 \times (r-1)}$ such that $(\mathbf{H}, \mathbf{1}_{n_1})^\top (\mathbf{H}, \mathbf{1}_{n_1})/n_1 = \mathbf{I}_r$. Let $r \ge 2$ and $\mathbf{U} \in \{-1,1\}^{n_1 \times (r-1)}$ with i.i.d. Rademacher entries. Then,*

$$\mathbb{P}\left\{\begin{array}{c} \|\mathbf{U}^\top \mathbf{H}/n_1\|_{op} \vee \|\mathbf{U}^\top \mathbf{1}_{n_1}/n_1\|_2 \le \sqrt{2\pi(r-1)/n_1} + \sqrt{8x/n_1} \\ \|\mathbf{U}^\top \mathbf{U}/n_1 - \mathbf{I}_{r-1}\|_{op} \le 4\sqrt{(3(r-1)+x)/n_1} \end{array}\right\} \ge 1 - 4e^{-x}.$$

*Suppose $n_1 p \le \sigma^2$. Let $\mu^2 = (n_1 p/\sigma^2)/20$. Then, for*

$$\varepsilon_n = \max\left\{\sqrt{40\pi r \sigma^2/(n_1^2 p)} + \sqrt{160 x \sigma^2/(n_1^2 p)}, 4\sqrt{(3r+x)/n_1}\right\},$$

$$\mathbb{P}\{\Delta_i' \vee \Delta_i'' \vee \Delta_i''' \le \varepsilon_n\} \ge 1 - 4e^{-x}.$$

*Proof.* Let $\mathbf{U} = (\mathbf{u}_1, \ldots, \mathbf{u}_{n_1})^\top$ and $\|\mathbf{v}\|_2 = 1$. As $\mathbb{E}(\mathbf{v}^\top \mathbf{u}_i)^{2m} \le \mathbb{E}(N(0,1))^{2m}$ for all $m$, for $t < 1/2$

$$\mathbb{E}\exp\big(t((\mathbf{v}^\top \mathbf{u}_i)^2 - 1)\big) \le \mathbb{E}\exp\big(t(N(0,1))^2 - 1)\big) \le \frac{e^{-t}}{(1-2t)^{1/2}} \le \exp\big(t^2/(1-2t)\big)$$

As $\mathbb{E}(1 - (\mathbf{v}^\top \mathbf{u}_i)^2)^2 = \mathbb{E}(\mathbf{v}^\top \mathbf{u}_i)^4 - 1 \le 2$,

$$\mathbb{E}\exp\big(t(1 - (\mathbf{v}^\top \mathbf{u}_i)^2)\big) \le 1 + 2(e^t - 1 - t) \le \exp\big(t^2/(1-2t)\big)$$

By the Bernstein inequality,

$$\mathbb{P}\big\{|\mathbf{v}^\top(\mathbf{I}_{r-1} - \mathbf{U}^\top \mathbf{U}/n_1)\mathbf{v}| \ge 2\sqrt{x/n_1} + 4x/n_1\big\} \le 2e^{-x}$$

Let $\varepsilon = 0.12$ and $N_\varepsilon \le (1 + 2/\varepsilon)^{r-1}$ be the $\varepsilon$-covering number for the unit ball in $\mathbb{R}^{r-1}$. We have

$$(1 - 2\varepsilon)\|\mathbf{U}^\top \mathbf{U}/n_1 - \mathbf{I}_{r-1}\|_{op} \le \max_{j \le N_\varepsilon} |\mathbf{v}_j(\mathbf{U}^\top \mathbf{U}/n_1 - \mathbf{I}_{r-1})\mathbf{v}_j|$$

with certain $\mathbf{v}_j$ with $\|\mathbf{v}_j\|_2 = 1$. Thus, as $1/(1 - 2\varepsilon) \leq 4/3$ and $\log(1 + 2/\varepsilon) \leq 3$,

$$\mathbb{P}\big\{\|\mathbf{U}^\top\mathbf{U}/n_1 - \mathbf{I}_{r-1}\|_{\mathrm{op}} \geq (8/3)\sqrt{(3(r-1)+x)/n_1} + 16(3(r-1)+x)/(3n_1)\big\} \leq 2e^{-x}.$$

When $4\sqrt{(3(r-1)+x)/n_1} < 1$, this implies

$$\mathbb{P}\big\{\|\mathbf{U}^\top\mathbf{U}/n_1 - \mathbf{I}_{r-1}\|_{\mathrm{op}} \geq 4\sqrt{(3(r-1)+x)/n_1}\big\} \leq 2e^{-x}.$$

Let $f(\mathbf{U}) = \|\mathbf{U}^\top\mathbf{H}/n_1^{1/2}\|_{\mathrm{op}}$. As $\mathbf{H}^\top\mathbf{H}/n_1 = \mathbf{I}_{r-1}$, $f(\cdot)$ is a unit-Lipschitz function, so that

$$\mathbb{P}\big\{f(\mathbf{U}) > \mathbb{E}f(\mathbf{U}) + t\big\} \leq e^{-t^2/8}.$$

Let $\mathbf{Z}$ be a standard Gaussian matrix. By the Sudakov-Fernique inequality

$$\mathbb{E}[|N(0,1)|]\mathbb{E}f(\mathbf{U}) \leq \mathbb{E}f(\mathbf{Z}) \leq 2\sqrt{r-1}$$

The proof is complete as the proof for $\mathbf{H}$ also applies with $\mathbf{H}$ is replaced by $\mathbf{1}_{n_1}$. $\qquad\square$

**Lemma 6.** *Let $\mathbf{M}_i$ be as in* (28), $\Delta_i' = \|\mathbf{U}_i^\top\mathbf{H}/(n_1\mu)\|_{op}$, $\Delta_i'' = \|\mathbf{U}_i^\top\mathbf{U}_i/n_1 - \mathbf{I}_{r-1}\|_{op}$ *and* $\Delta_i''' = \|\mathbf{U}_i^\top\mathbf{1}_{n_1}/(n_1\mu)\|_2$. *Then, the $r$-th singular value of $\mathbf{M}_i$ is bounded by $\sigma_r(\mathbf{M}_i) \geq (p/10)\sqrt{n_1 2k_2(1 - 2\mu^2\Delta_i' - \mu^2\Delta_i'' - (1 + 1/92)\mu^4(\Delta_i''')^2)_+}$.*

*Proof.* Write $\overline{\mathbf{H}} = (\mathbf{H}, -\mathbf{H})$, $\mathbf{M}_1 = \sqrt{1 - \mu^2}\,\overline{\mathbf{H}} + 5\mathbf{1}_{n_1 \times (2r-2)}$ and $\overline{\mathbf{U}}_i = (\mathbf{U}_i, -\mathbf{U}_i)$. We have

$$\sigma_r^2(\mathbf{M}_i)/n_1 = \sigma_r(\mathbf{M}_i^\top\mathbf{M}_i)/n_1 \geq k_2(p/10)^2\sigma_r\big((\mathbf{M}_1 + \mu\overline{\mathbf{U}}_i)^\top(\mathbf{M}_1 + \mu\overline{\mathbf{U}}_i)/n_1\big).$$

Let $\overline{\mathbf{I}}_{r-1} = (\mathbf{I}_{r-1}, -\mathbf{I}_{r-1})$ and $\overline{\mathbf{u}}_i = \overline{\mathbf{U}}_i^\top\mathbf{1}_{n_1}/n_1$. As $\|\overline{\mathbf{U}}_i^\top\overline{\mathbf{U}}_i/n_1 - \overline{\mathbf{I}}_{r-1}^\top\overline{\mathbf{I}}_{r-1}\|_{\mathrm{op}} = 2\Delta_i''$,

$$\begin{aligned}
&\sigma_r\big((\mathbf{M}_1 + \mu\overline{\mathbf{U}}_i)^\top(\mathbf{M}_1 + \mu\overline{\mathbf{U}}_i)/n_1\big)\\
&\geq \sigma_r\big(\mathbf{M}_1^\top\mathbf{M}_1/n_1 + \mu^2\overline{\mathbf{U}}_i^\top\overline{\mathbf{U}}_i/n_1 + 5\mu\overline{\mathbf{u}}_i\mathbf{1}_{2r-2}^\top + 5\mu\mathbf{1}_{2r-2}\overline{\mathbf{u}}_i^\top\big)\\
&\quad - \mu\|\overline{\mathbf{U}}_i^\top\overline{\mathbf{H}}/n_1 + \overline{\mathbf{H}}^\top\overline{\mathbf{U}}_i/n_1\|_{\mathrm{op}}\\
&\geq \sigma_r\big(\mathbf{M}_1^\top\mathbf{M}_1/n_1 + \mu^2\overline{\mathbf{I}}_{r-1}^\top\overline{\mathbf{I}}_{r-1} + 5\mu\overline{\mathbf{u}}_i\mathbf{1}_{2r-2}^\top + 5\mu\mathbf{1}_{2r-2}\overline{\mathbf{u}}_i^\top\big) - 2\mu^2\Delta_i'' - 4\mu^2\Delta_i'
\end{aligned}$$

by Weyl's inequality.

Assume $\|\overline{\mathbf{u}}_i\|_2 = \sqrt{2}\mu\Delta_i''' > 0$. As $\mathbf{M}_1^\top\mathbf{M}_1/n_1 = (1 - \mu^2)\overline{\mathbf{I}}_{r-1}^\top\overline{\mathbf{I}}_{r-1} + 25\mathbf{1}_{(2r-2)\times(2r-2)}$,

$$\begin{aligned}
&\mathbf{M}_1^\top\mathbf{M}_1/n_1 + \mu^2\overline{\mathbf{I}}_{r-1}^\top\overline{\mathbf{I}}_{r-1} + 5\mu\overline{\mathbf{u}}_i\mathbf{1}_{2r-2}^\top + 5\mu\mathbf{1}_{2r-2}\overline{\mathbf{u}}_i^\top\\
&= \overline{\mathbf{I}}_{r-1}^\top\overline{\mathbf{I}}_{r-1} - \frac{2\overline{\mathbf{u}}_i\overline{\mathbf{u}}_i^\top}{\|\overline{\mathbf{u}}_i\|_2^2} + \left(\frac{\mathbf{1}_{2r-2}}{\sqrt{2r-2}}, \frac{\overline{\mathbf{u}}_i}{\|\overline{\mathbf{u}}_i\|_2}\right)\begin{pmatrix} B & \sqrt{B\varepsilon} \\ \sqrt{B\varepsilon} & 2 \end{pmatrix}\left(\frac{\mathbf{1}_{2r-2}}{\sqrt{2r-2}}, \frac{\overline{\mathbf{u}}_i}{\|\overline{\mathbf{u}}_i\|_2}\right)^\top
\end{aligned}$$

with $B = 25(2r-2) \geq 50$ and $\varepsilon = \mu^2\|\overline{\mathbf{u}}_i\|_2^2 = 2\mu^4(\Delta_i''')^2$. As $\overline{\mathbf{I}}_{r-1}^\top\overline{\mathbf{I}}_{r-1}/2$ is an orthogonal projection with $\overline{\mathbf{u}}_i/\|\overline{\mathbf{u}}_i\|_2$ as an eigenvector, the $r$-th eigenvalue of the above matrix is

$$\sigma_r' = \big(B + 2 - \sqrt{(B+2)^2 - 4(2B - B\varepsilon)}\big)/2.$$

For $\varepsilon \leq 1$, $\sqrt{(B+2)^2 - 2B(4 - 2\varepsilon)} = \sqrt{(B - 2 + 2\varepsilon)^2 + 4(2\varepsilon - \varepsilon^2)} \leq B - 2 + \varepsilon + 4\varepsilon/46$, which implies

$$\sigma_r' \geq \frac{2B(2-\varepsilon)}{B + 2 + B - 2 + \varepsilon + 4\varepsilon/46} \geq (2-\varepsilon)(1 - (25/46)\varepsilon/B) \geq 2 - (1 + 1/92)\varepsilon.$$

Hence, the conclusion holds. The conclusion holds automatically when $\varepsilon > 1$. The proof for $\varepsilon = 0$ is simpler and omitted. $\qquad\square$