# OpenReview forum: "Rate-Optimal Subspace Estimation on Random Graphs"
_NeurIPS.cc/2021/Conference — NeurIPS 2021 Poster_

### Official Review · Reviewer_MzEw · 2021-07-15

**Rating:** 6
**Confidence:** 4

**Summary:**

The authors study the problem of estimating the connectivity matrix $M \in [0,1]^{n_1 \times n_2}$ of a random bipartite graph (with $n_1 \times n_2$ vertices) given independent observations $A_{ij} \sim Bernoulli(M_{ij})$ for $(i,j) \in [n_1] \times [n_2]$. Assuming M is low rank, they consider algorithms based on singular value thresholding (Algs. 1, 2) and derive estimation error bounds (in the operator and Frobenius norms) in Theorems 1, 2. These bounds are shown to be optimal in Theorem 3. Moreover, the bounds in Theorem 1 are then used to bound the error in estimating the column space of M (Theorem 4), which are shown to be optimal in Theorem 5.

**Limitations And Societal Impact:**

The authors have addressed the limitations of the results in Section 5.

**Main Review:**

The paper studies an interesting problem which is important in practice since bipartite graphs appear in many applications. The results are in general presented in a clean manner. However, the quality of writing really needs to be improved throughout since there are quite a few grammatical mistakes. I also have some detailed remarks which I have listed below.

(1)	The regularization scheme used in Algos. 1, 2 is quite standard in the literature and is used, for instance, to show concentration of sparse inhomogeneous Erdos Renyi graphs [1]. It will be helpful to discuss this so that the reader has better perspective.

(2)	Theorem 1 is under the setting $n_1 \geq n_2$ and it is mentioned at the beginning of Section 2 that this is w.l.o.g. Also, theorem 4 is written in a more general form without this assumption so it might be better to do the same for Theorem 1 as well, for consistency. It wasn’t clear to me why the lower bound in Theorem 5 requires $n_1 \geq n_2$, and although there is a discussion in Section 5, some additional discussion would be helpful.

(3)	Currently, there is no sketch of the proof ideas in the main text for any of the theorems. I made a brief pass of the appendix to go through the steps, but it will be good to add the main ideas within the main text.

(4)	In Section 3, the error definitions in lines 160,161 could be removed since eqs. (10), (11) clearly define the subspace error terms (these are standard). The bound in Theorem 4 should follow in a straightforward manner from Theorem 1 using the Davis-Kahan theorem or Wedin’s bound, the latter of which is specifically for Singular values/subspaces.

(5)	In Section 3.2, isn’t the definition of $P_M$ simply $U U^T$? The statement of Theorem 5 also seems to have an inaccuracy in the stated conditions since it requires $C_0 r \leq \frac{n_1 n_2p^2}{\sigma_{*}^2} \leq (5/2)^2$ which will typically not be satisfied.

(6)	In the experiments, there are currently no comparisons with other methods from the literature for similar low-rank matrix estimation problems. This would have helped us see how other methods fare in simulations against the proposed methods.

References:

[1] Le, C.M., Levina, E. and Vershynin, R. (2017), Concentration and regularization of random graphs. Random Struct. Alg., 51: 538-561.

---------------------- Post author response -------------------------

I am reasonably satisfied with the response of the authors to my comments, and I will increase my score to 6. I hope the authors will provide a more detailed discussion on the significance of their results in relation to those for related problems in the literature (especially bipartite SBMs).

**Time Spent Reviewing:**

around 12 hours

---

> ### Author Response · Authors · 2021-08-11
> **Response to Reviewer MzEw**
>
> Thank you very much for your comments.
>
> (1) The regularization scheme used in Algos. 1, 2 is quite standard in the literature and is used, for instance, to show concentration of sparse inhomogeneous Erdos Renyi graphs [1]. It will be helpful to discuss this so that the reader has better perspective.
>
> Thank you for your suggestion. We have discussed the details about regularization in Remark 1 and 2. In Remark 2, we present the error rate the algorithm can obtain without regularization. In Remark 2, we also cite the paper [2], which is the earliest result about regularized graph we found. The author of [1] provides an alternative proof of the existing result. In the revised version, we will add further discussion on concentration of sparse inhomogeneous Erdos-Renyi graphs and why it can improve the error rate.
>
> (2) Theorem 1 is under the setting  and it is mentioned at the beginning of Section 2 that this is w.l.o.g. Also, theorem 4 is written in a more general form without this assumption so it might be better to do the same for Theorem 1 as well, for consistency. It wasn’t clear to me why the lower bound in Theorem 5 requires, and although there is a discussion in Section 5, some additional discussion would be helpful.
>
> Thank you for your suggestion. We will modify the notation in Theorem 1. More precisely,  we just need to replace $n_1$ by $n_1\vee n_2$ if we do not assume $n_1\ge n_2$.
>
> (3) Currently, there is no sketch of the proof ideas in the main text for any of the theorems. I made a brief pass of the appendix to go through the steps, but it will be good to add the main ideas within the main text.
>
> Thank you for your suggestion. We will add the main ideas to the main text. We have sketched the proof ideas of Theorem 4 in line 179. The ideas of proving Theorem 5 cannot be explained in a few sentences. We will add a subsection the describe the main ideas in the final version.
>
> (4) In Section 3, the error definitions in lines 160,161 could be removed since eqs. (10), (11) clearly define the subspace error terms (these are standard). The bound in Theorem 4 should follow in a straightforward manner from Theorem 1 using the Davis-Kahan theorem or Wedin’s bound, the latter of which is specifically for Singular values/subspaces.
>
> Thank you for your suggestion. In lines 160 and 161, we define a more general loss function, which does not assume $U_i$'s are orthonormal. The term $(U^TU)^{-1}$ is required in this definition. If we consider singular vector, which are orthonormal by default, then we do not need $(U^TU)^{-1}$. To simplify our presentation of the definition, we will assume $U_i$'s are orthonormal and only keep the definition in eqs. (10) and (11).
>
> (5) In Section 3.2, isn’t the definition of $P_M$ simply $UU^T$?
>
> Yes. Since we assume $U$ has orthonormal columns, we can simply define it in this way. We will revise the definition of $P_M$.
>
> (6) The statement of Theorem 5 also seems to have an inaccuracy in the stated conditions since it requires $C_0r\le \frac{n_1n_2p^2}{\sigma_*^2}\le (5/2)^2$ which will typically not be satisfied.
>
> To construct this lower bound, we consider the case $r=1$ and case $r>1$ separately.
>
> (a) If $r=1$, we assume $n_1n_2(4p/10)^2\le r\sigma_*^2\le n_1n_2(6p/10)^2$.
>
> (b) If $r>1$, we assume, $r\sigma_*^2\le n_1n_2p^2/C_0$.
>
> Since $r=1$ and $r>1$ will not happen simultaneously, it does not require $C_0r\le \frac{n_1n_2p^2}{\sigma_*^2}\le (5/2)^2$. We hope this can resolve your concern. We believe the statement of this theorem has been accurately presented, but we will rephrase the assumptions to avoid the confusion.
>
> The reasons for why assumption (a) are needed has been discussed in Remark 4.
>
>
> (7) In the experiments, there are currently no comparisons with other methods from the literature for similar low-rank matrix estimation problems. This would have helped us see how other methods fare in simulations against the proposed methods.
>
> Existing methods for low-rank matrix estimation can be divided into two types: hard thresholding [3,4] and soft thresholding [5,6]. We have present both types of methods in our Algorithm 1 and 2. The existing algorithms has not consider the special property of sparse random graphs. The comparison of algorithms with and without regularization appears in Section 4.1. We have two important observations from our experiments:
>
> (a) Given a "good" regularization, hard thresholding method performs better than soft thresholding method.
>
> (b) Regularization has less effect on soft thresholding.
>
> Therefore, we believe we have already compared the performance of existing algorithms and test how regularization affect the estimation error. We will explain the connection between our experiments and existing methods with more details.
>
> [1] Le, C.M., Levina, E. and Vershynin, R. (2017), Concentration and regularization of random graphs. Random Struct. Alg., 51: 538-561.
>
> [2] Uriel Feige and Eran Ofek. Spectral techniques applied to sparse random graphs. Random
> Structures \& Algorithms, 27(2):251–275, 2005.
>
> [3] Sourav Chatterjee et al. Matrix estimation by universal singular value thresholding. The Annals of Statistics, 43(1):177–214, 2015.
>
> [4] T Tony Cai, Anru Zhang, et al. Rate-optimal perturbation bounds for singular subspaces with
>  applications to high-dimensional statistics. The Annals of Statistics, 46(1):60–89, 2018.
>
> [5] Vladimir Koltchinskii, Karim Lounici, Alexandre B Tsybakov, et al. Nuclear-norm penalization and  optimal  rates  for  noisy  low-rank  matrix  completion.The  Annals  of  Statistics,  39(5):3062302–2329, 2011
>
> [6] Jian-Feng Cai, Emmanuel J Candès, and Zuowei Shen. A singular value thresholding algorithm for matrix completion.SIAM Journal on optimization, 20(4):1956–1982, 2010.

---

### Official Review · Reviewer_vRfR · 2021-07-15

**Rating:** 6
**Confidence:** 3

**Summary:**

The authors discuss the estimation of a connectivity matrix M \in [0,p]^{n_1 \times n_2} based on one instance of a generated random bipartite graph where node i of the one part is connected with node j of the other part with probability M_{i,j}. The authors discuss estimation of M in terms of operator and Frobenius norm, as we all estimation of the singular space of M.

The authors provide tight minimax results for all three notions of recovery. Their upper bounds are through a relatively (regularized) simple singular value thresholding (SVT) algorithm, not far from the well-known in the field universal singular value thresholding algorithm (USVT) by Chaterjee.

**Ethical Concerns:**

I see no ethical concern.

**Limitations And Societal Impact:**

I mentioned the limitations above. I see no potential negative societal impact.

**Main Review:**

## Originality

I think the results in this work are original, but I could be missing some part of the literature. The results are definitely naturally highly overlapping with results/methods from literature of estimating SBMs and graphons but yet offer a new insight that I have not encountered before because of the bipartite structure.

That being said, the bipartite structure appears to make many things more tractable from a mathematical point of view. This, for example, is obvious in the lower bounds for the Frobenius/operators norms (Appendix C) where the construction is far easier than similar lower bound construction in the context of graphons (see e.g. (Klopp, Tsybakov, Verzelen : Oracle inequalities for network models and sparse graphon
estimation, 2015) and (Mcmillan, Smith: When is nontrivial estimation possible for Graphons and Stochastic Block Models?, 2016)).

## Quality/Clarity

The paper is of good quality, the statements are clean and the proofs as far as I checked are clear and make sense. One aspect of the results that confused me it is that the lower bound for the singular space estimation (Theorem 5) is a lower bound in expectation, while the upper bound (Theorem 4), please explain more on why this is the case and how/whether it can be resolved.
Presentation-wise, one aspect of the result that would help is an interpretation of the rates. For example, a natural rate appearing in parameter estimation problems is the so-called parametric rate (see for example the discussion in (Gao, Lu, Zhou Rate-optimal graphon estimation, 2014). My understanding is that in the case of the Frobenius norm the rate n_1pr indeed corresponds to the parametric rate and that explains a lot. Also, the dependence on p is not as surprising, as it is similar to the one appearing in Klopp, Tsybakov, Verzelen (2015) and Mcmillan, Smith (2016) for SBMs.

## Significance

If I am correct that the results are accurate, I think the results are significant. There is a large literature in rate-optimal estimation of various graphon structures, but the simpler bipartite case seems to be missing. The authors close this gap. Yet, I really think the authors should work more on positioning their paper in the literature.


## Missing literature

The authors seem to miss some important references that are similar to the present work, and a deep comparison between the results is necessary to appreciate the results. For a dense setting, (Gao, Lu, Zhou Rate-optimal graphon estimation, 2014) establish the tight rate for SBM with r blocks. This corresponds to the case where p=constant in the submitted paper. Of course in the submitted paper the model is not an SBM and it is bipartite. But how do the results compare? How are they different?

Similarly, the sparse case has been studied for SBM by Klopp, Tsybakov, Verzelen (2015) and Mcmillan, Smith (2016)). Please compare the results and highlight the significance and new insights.

**Time Spent Reviewing:**

2

---

> ### Author Response · Authors · 2021-08-11
> **Response to Reviewer vRfR**
>
> Thank you very much for your comments.
>
> 1. Comparison with stochastic block models (SBM).
>
> We believe our parameter space is very different from (bipartite) SBM.
>
> To simplify our discussion, we let $n=n_1=n_2$. The number of blocks in SBM determines the rank of connectivity matrix $\mathbb E[A]$ in SBM, so we let $r=k$ be number of blocks.
>
> In the SBM setting, the probability of connection depends on the membership of the nodes, so there are $r^2$ many parameters in the connectivity matrix. Suppose the assumptions of the SBM allow exact recovery of the blocks (see [5] for the conditions), then, on average, we can use $n^2/r^2$ many Bernoulli samples to estimate one parameter.
>
> In our setting, the only assumption on the connectivity matrix is "low-rank", so the number of parameters has order $O(nr)$.  On average, we use $n^2/(nr) = n/r$ many Bernoulli samples to estimate one parameter. Thus, this model is more "difficult" than SBM since we need to estimate more parameters.
>
> Therefore, the result in SBM cannot be directly applied to our problem. In contrast, the lower bounds we proved in the paper cannot be applied to SBM, since SBM is a strictly smaller model.
>
> Since SBM is a low-rank model, spectral methods have been applied to SBM. The discussion appears at the end of Section 2.2. Of course, we will add more references and comparison with SBM in the final version.
>
> 2. Comparison with graphon models.
>
> In the proof of minimax rate of estimating graphon model [1,2], the authors use SBM to estimate a graphon, where the number of the blocks depends on the smoothness assumption of the graphon. Because of the difference between SBM and our low-rank model discussed in our manuscript, it is also difficult the compare the graphon model and our problem.
>
> In another paper [3], spectral methods are applied to graphon estimation, but they fail to achieve the optimal rate. It is worth noting that estimating graphon by SBM is not computationally feasible, while spectral methods can be implement in polynomial-time. The result of this paper may partially explain the difference and connection among low-rank model, SBM and graphon model.
>
> In Section 3 of our paper, we focus on singular space estimation. This problem is never consider in graphon model since the there is no low-rank structure in a general graphon model.
>
> 3. Similarity with other lower bounds.
>
> We have pointed out at the end of Section 2.3 that the bounds in Theorem 3 have been considered in [4], while our bound is tighter. Indeed, the paper [4] appears earlier than the results about SBM and graphons. It is true that the lower bounds in SBM or graphon model is more complicated to established than the one of low-rank model.
>
> However, we believe the lower bound in Theorem 5 is novel. Although the lower bounds for estimating subspaces has been considered in literature, the bounded entries in random graph models make the analysis of the lower bounds quite nontrivial.
>
> 4. Lower bounds in Theorem 5.
>
> This is a technical issue and we can resolve it. In our current proof, we define a prior on the parameter subspace, then minimax lower bound can be given by the Bayesian error. We believe the current version of the presentation of the proof is easier for reader to follow. We can obtain the Fano's type lower bound by applying Fano's inequality on the parameter subspace.
>
> [1] Gao, Chao, Yu Lu, and Harrison H. Zhou. "Rate-optimal graphon estimation." The Annals of Statistics 43.6 (2015): 2624-2652.
>
> [2] Klopp, Olga, Alexandre B. Tsybakov, and Nicolas Verzelen. "Oracle inequalities for network models and sparse graphon estimation." The Annals of Statistics 45.1 (2017): 316-354.
>
> [3] Xu, Jiaming. "Rates of convergence of spectral methods for graphon estimation." International Conference on Machine Learning. PMLR, 2018.
>
> [4] Vladimir Koltchinskii, Karim Lounici, Alexandre B Tsybakov, et al. Nuclear-norm penalization and  optimal  rates  for  noisy  low-rank  matrix  completion.The  Annals  of  Statistics,  39(5):3062302–2329, 2011
>
> [5] Abbe, Emmanuel. "Community detection and stochastic block models: recent developments." The Journal of Machine Learning Research 18.1 (2017): 6446-6531.

---

> > ### Comment · Reviewer_vRfR · 2021-08-27
> > **Thanks for the response**
> >
> > I thank their reviewer for their response, it will be good to add in the revised version the suggested literature for a more complete literature review. It seems also good to resolve the technical issue of Theorem 5.

---

### Official Review · Reviewer_f1zZ · 2021-07-19

**Rating:** 6
**Confidence:** 3

**Summary:**

This paper considers the problem of learning a random bipartite graph with sparse connectivity matrix M. They show that the performance of estimating the matrix M depends on the sparsity, and focus on two types of performance: the error of estimating M and the error of estimating the column space of M. They show that the estimators achieve the minimax optimal rate.

**Limitations And Societal Impact:**

Yes

**Main Review:**

The result of this paper is novel in that the authors uses algorithms similar to universal singular value thresholding and soft singular value thresholding, but provide the minimax lower bounds for the error estimation. I think the singular space estimation bounds are of great value to the statistics community, however for the larger NeurIPS audience, I find that more applications and experiments on applications, (such as clustering), should be discussed. Also, the paper is most probably confusing to read as the problem statement is not written clearly, hence it would be better if that was done, as well as a clear set of main contributions to be written out in the introduction of the paper in point form, that is distinct from the literature review.

**Time Spent Reviewing:**

7

---

> ### Author Response · Authors · 2021-08-11
> **Response to Reviewer f1zZ**
>
> Thank you very much for your comments.
>
> 1. More applications and experiments on applications, (such as clustering), should be discussed.
>
> Thank you very much for your suggestion. In our model, there is no cluster structure in the parameter space. However, we explain the connection of our result to stochastic block model in the last paragraph of Section 2.2.
>
> 2. A clear set of main contributions should be written out in the introduction of the paper in point form.
>
> Thank you for the suggestion. We have described our main contributions in the paragraphs starting from line 29 and the paragraph starting from 44. In a later version, we will list them in a point form.

---

### Official Review · Reviewer_iUnH · 2021-07-29

**Rating:** 6
**Confidence:** 4

**Summary:**

This paper considers the problem of estimating bipartite graphs with low rank structures. The authors develop several simple estimators for estimating the connectivity matrix and its column and row space, based on singular value thresholding. The optimal rates are established by deriving the matching minimax lower bounds and upper bounds.

**Limitations And Societal Impact:**

There are several points listed to be clarified.

1. The authors should make clear the motivation of estimating the connectivity matrix of a bipartite graph, as well as its column and row space. In particular, it is unclear the motivation of the setting of the problem considered, i.e., Eq. (1). Does there exist similar problem setting in the literature? Is the reason why the authors focus on bipartite graphs just that its connectivity matrix can be represented by an n_1 \times n_2 matrix?

2. The minimax lower bound and upper bound are usually provided in the expectation form. The authors are suggested to provide the bounds of the expectation of the estimation error in theorems.

3. There are four parameters, n_1, n_2, p, and r, in constructing the parameter space. The authors should clarify clearly the dependence of those parameters with each other in establishing the optimal rate, to make sure that all the dependence conditions are compatible.

4. In Theorem 3, the authors should provide the lower bound of n_1, but not just claim that n_1 is sufficiently large.

5. In Theorem 5, the authors claim that the constant C_0 should be sufficiently large. What is the condition on C_0? Is it compatible with another condition stated in the theorem, r \leq n_1 / C_0?

6. It is observed that some numbers stated in Theorem 3 are not consistent with the ones in its proof. In addition, the authors should provide more details in proving Lemma 1.



**Main Review:**

The main contributions of this paper focus on theoretical parts. More specifically, the authors establish the optimal rates for estimating the connectivity matrix and its column and row space. The minimax optimal rate is a fundamental problem in parameter estimation. To achieve this, the upper bound is derived by designing some estimators based on singular value thresholding. The minimax lower bound is obtained by applying Fano’s inequality in a subset of the parameter space.


**Time Spent Reviewing:**

6

---

> ### Author Response · Authors · 2021-08-11
> **Response to Reviewer iUnH**
>
> Thank you very much for your comments.
>
>
> 1. Does there exist similar problem setting in the literature?
>
>
> Yes, there are previous works considering the same model assumptions. For example, the authors in [1] consider a similar problem. The result in [2] also focus on random bipartite graph. We also cite other papers working the applications of our model, e.g., community detection [6, 7].
>
> 2. The authors are suggested to provide the bounds of the expectation of the estimation error in theorems.
>
> Thank you for pointing this out. We believe both ways to present minimax error appear widely in literature. For example, high probability bounds appear in [1,2,4,9] and expectation of estimation error is presented in [3, 8].
>
> 3. There are four parameters, $n_1, n_2, p$, and $r$, in constructing the parameter space. The authors should clarify clearly the dependence of those parameters with each other in establishing the optimal rate, to make sure that all the dependence conditions are compatible.
>
> In Theorem 1 to 4, the dependence on the parameters are very mild. Generally speaking, we only require the size of the random bipartite graph to be sufficiently large.
>
> The assumptions become trickier when we have the gap of singular value $\sigma_*$. We have discuss the compatibility in equation (8). Because $\sigma^*$ is the smallest nonzero singular value of a rank-$r$ matrix, so there is an upper bound for $\sigma^*$ depending on other parameters. More details appear in the first paragraph of Section 3.
>
> In Theorem 5, we need more assumptions on the parameters. However, it is clear that those assumptions do not violate the constrained in equation (8). More discussion about these assumptions can be found in Remark 4.
>
> 4. In Theorem 3, the authors should provide the lower bound of $n_1$, but not just claim that $n_1$ is sufficiently large. In Theorem 5, the authors claim that the constant $C_0$ should be sufficiently large. What is the condition on $C_0$?
>
> Thanks for pointing this out. We will present these conditions with numerical values for the lower bounds on $n_1$ and $C_0$. However, many absolute constants will not be presented in literature, please see the theorems in [1,2,4,5,9]. We believe the current version of the theorems will not affect the correctness and applicability.
>
> 5. Is $C_0$ sufficiently large compatible with another condition stated in the theorem, $r \leq n_1 / C_0$?
>
> Yes, it is compatible. This condition means the size of the matrix is sufficiently larger than the rank of the matrix. In other words, we focus on low rank connectivity matrix.
>
> 6. It is observed that some numbers stated in Theorem 3 are not consistent with the ones in its proof.
>
> Thanks for your careful review. We will correct those numerical values in the final version.
>
> 7. In addition, the authors should provide more details in proving Lemma 1.
>
> We believe the proof of Lemma 1 is complete. However, we will add more details and intuition in the final version. Thanks for letting us know more details are needed in this proof.
>
> [1] Vladimir Koltchinskii, Karim Lounici, Alexandre B Tsybakov, et al. Nuclear-norm penalization and  optimal  rates  for  noisy  low-rank  matrix  completion.The  Annals  of  Statistics,  39(5):3062302–2329, 2011
>
> [2] Mark A Davenport,  Yaniv Plan,  Ewout Van Den Berg,  and Mary Wootters.   1-bit matrix completion.Information and Inference: A Journal of the IMA, 3(3):189–223, 2014.
>
> [3] T Tony Cai, Anru Zhang, et al. Rate-optimal perturbation bounds for singular subspaces with
>  applications to high-dimensional statistics. The Annals of Statistics, 46(1):60–89, 2018.
>
> [4] Sourav Chatterjee et al. Matrix estimation by universal singular value thresholding. The Annals of Statistics, 43(1):177–214, 2015.
>
> [5] Can M Le, Elizaveta Levina, and Roman Vershynin. Concentration and regularization of random
>  graphs. Random Structures \& Algorithms, 2017.
>
> [6] Mohamed Ndaoud, Suzanne Sigalla, and Alexandre B Tsybakov. Improved clustering algorithms for the bipartite stochastic block model. arXiv preprint arXiv:1911.07987, 2019.
>
> [7] Zhixin Zhou and Arash A Amini. Analysis of spectral clustering algorithms for community
> detection: the general bipartite setting. Journal of Machine Learning Research, 20(47):1–47,
> 2019.
>
> [8] Tony Cai and Wen-Xin Zhou. A max-norm constrained minimization approach to 1-bit matrix
> completion. The Journal of Machine Learning Research, 14(1):3619–3647, 2013.
>
> [9] Changxiao Cai, Gen Li, Yuejie Chi, H Vincent Poor, and Yuxin Chen. Subspace estimation from
>  unbalanced and incomplete data matrices: $\ell_{2,\infty}$ statistical guarantees. The Annals of Statistics, 49(2):944–967, 2021.

---

### Decision · Program_Chairs · 2021-09-27

**Decision:**

Accept (Poster)

**Comment:**

The reviewers gave very coherent marks to this paper, generally recognizing its quality and the value of its contribution, in particular in filling the gap of identifying minimax optimal estimators for estimation of an average connectivity matrix and of its column space from an observed bipartite random graph with M as its expected adjacency matrix.
The paper thus appears well suited for acceptance. We recommend that the authors take into account the reviewers' comments especially about positioning their results when preparing the final version.